# Id4 promotes the elimination of the pro-activation factor Ascl1 to maintain quiescence of adult hippocampal stem cells

Isabelle Maria Blomfield[1], Brenda Rocamonde[2], Maria del Mar Masdeu[1], Eskeatnaf Mulugeta[3], Stefania Vaga[1], Debbie LC van den Berg[3], Emmanuelle Huillard[2], François Guillemot[1]*, Noelia Urbán[1,4]*

[1]The Francis Crick Institute, London, United Kingdom; [2]Institut du Cerveau et de la Moelle Epinière, ICM, Inserm U 1127, CNRS UMR 7225, Sorbonne Université, Paris, France; [3]Department of Cell Biology, Erasmus MC, Rotterdam, Netherlands; [4]Institute of Molecular Biotechnology (IMBA), Vienna Biocenter Campus (VBC), Vienna, Austria

**Abstract** Quiescence is essential for the long-term maintenance of adult stem cells but how stem cells maintain quiescence is poorly understood. Here, we show that neural stem cells (NSCs) in the adult mouse hippocampus actively transcribe the pro-activation factor Ascl1 regardless of their activated or quiescent states. We found that the inhibitor of DNA binding protein Id4 is enriched in quiescent NSCs and that elimination of Id4 results in abnormal accumulation of Ascl1 protein and premature stem cell activation. Accordingly, Id4 and other Id proteins promote elimination of Ascl1 protein in NSC cultures. Id4 sequesters Ascl1 heterodimerization partner E47, promoting Ascl1 protein degradation and stem cell quiescence. Our results highlight the importance of non-transcriptional mechanisms for the maintenance of NSC quiescence and reveal a role for Id4 as a quiescence-inducing factor, in contrast with its role of promoting the proliferation of embryonic neural progenitors.
DOI: https://doi.org/10.7554/eLife.48561.001

*For correspondence:
francois.guillemot@crick.ac.uk
(FG);
noelia.urban@imba.oeaw.ac.at
(NU)

## Introduction

Tissue stem cells must maintain their long-term activity while minimising the accumulation of genetic and metabolic damages. In several adult tissues, stem cells can remain inactive for long periods of time in a state of quiescence. Specific stimuli promote the exit from quiescence of different types of adult stem cells, such as hypoxia for stem cells of the carotid body or muscle injury for satellite stem cells (*Dumont et al., 2015*; *Sobrino et al., 2018*). Regulation of the transit between quiescent and active compartments is essential to maintain a pool of stem cells able to sustain tissue homeostasis and provide an adequate response to insults over the lifespan of the organism. An excessive retention of stem cells in the quiescent compartment would not produce enough differentiated progeny to maintain functionality, as happens for instance during aging (*García-Prat et al., 2016*; *Leeman et al., 2018*). On the other hand, excessive stem cell activity would eventually result in stem cell exhaustion, also leading to loss of functionality (*Castilho et al., 2009*; *Ho et al., 2017*). Quiescence is an essential property of cancer stem cells that allows them to evade immune surveillance and results in resistance to treatment (*Agudo et al., 2018*). Despite their relevance for the fields of tissue repair, aging and cancer biology, the mechanisms regulating quiescence in adult stem cells are still largely unknown.

**eLife digest** Stem cells in embryos give rise to all the tissues in the body. Adults also have stem cells, but they are fewer in number and they are usually dedicated to repairing and regenerating specific tissues. A region of the brain called the hippocampus, which is involved in learning, memory and mood, has a pool of neural stem cells. These cells can produce new brain cells long into adulthood, but maintaining their regenerative potential is a balancing act. Enough new brain cells need to be made to keep up with the brain's demands, but if every stem cell matured into a brain cell, the brain's capacity for repair would be lost. So, some neural stem cells hit a metaphorical snooze button to enter a resting state known as quiescence.

Stem cells in the hippocampus make a protein called Ascl1 that interacts with DNA to switch on quiescent cells so they will divide and mature. Left unchecked, Ascl1 could deplete the stem cell supply, so resting stem cells must have a way to turn Ascl1 off, but it was previously unknown how. Clues point to the E proteins, which interact with Ascl1 to allow it to bind to DNA. If the E proteins are not present, Ascl1 cannot work as a genetic switch. E proteins can also interact with inhibitor of DNA binding/differentiation proteins, known as Id proteins for short. To find out whether Id proteins affect Ascl1 activity, Blomfield et al. looked at stem cells in the hippocampus of adult mice, and at quiescent stem cells grown in the laboratory.

Blomfield et al. showed that all stem cells in the hippocampus make Ascl1, but its levels are much lower when stem cells are resting. This difference was down to an Id protein called Id4. In resting stem cells, Id4 interacted with E proteins, preventing them from binding to Ascl1, and stopping Ascl1 from 'waking up' the cells. This not only left Ascl1 unable to activate its target genes, it also made it vulnerable to destruction by the cell's protein recycling system. Mice with no Id4 in their hippocampus stem cells had higher levels of Ascl1, and their stem cells were more active.

The number of stem cells in a resting state increases as we age, and in illnesses like depression, limiting brain cell replacement. Uncovering the signals that switch Id4 on or off could reveal why stem cells rest more with age and illness. This could help us find ways to kick-start the production of new brain cells in adulthood.

DOI: https://doi.org/10.7554/eLife.48561.002

In the adult brain, NSC populations in the ventricular-subventricular zone (V-SVZ) of the lateral ventricles and in the dentate gyrus (DG) of the hippocampus, generate new neurons and glia that integrate into pre-existing neuronal networks (*Bond et al., 2015*; *Lim and Alvarez-Buylla, 2016*). In both regions, a large fraction of stem cells is quiescent. Extensive work has led to the identification of extracellular signals present in the V-SVZ and DG niches that regulate quiescent and active states (*Choe et al., 2016*; *Silva-Vargas et al., 2013*). Notch, BMP4 and the neurotransmitter GABA have been shown to maintain stem cell quiescence while Wnt, Shh and the neurotransmitter glutamate are thought to promote stem cell activity (*Bao et al., 2017*; *Choe et al., 2016*; *Engler et al., 2018*; *Imayoshi et al., 2010*; *Lie et al., 2005*; *Mira et al., 2010*; *Petrova et al., 2013*; *Qu et al., 2010*; *Song et al., 2012*; *Yeh et al., 2018*). In contrast, little is known of the cell intrinsic machinery that NSCs employ to adjust their activity to the different signals received from the niche. Ascl1 is one of the few intrinsic regulators of NSC quiescence described so far. Ascl1 is a basic-helix-loop-helix (bHLH) transcription factor that is present in a fraction of dividing stem cells and intermediate progenitors in the adult hippocampus. Loss of Ascl1 completely blocks the activation of adult hippocampal stem cells, inhibits the generation of new neurons and prevents the depletion of the stem cell pool over time (*Andersen et al., 2014*). Ascl1 may therefore determine the balance between quiescence and activity of hippocampal NSCs. Indeed, stabilization of Ascl1 protein by inactivation of the E3 ubiquitin ligase Huwe1 results in over-proliferation of hippocampal stem cells and prevents their return to quiescence (*Urbán et al., 2016*). However, Huwe1 inactivation is not sufficient to trigger the large-scale activation of quiescent stem cells, indicating that additional mechanisms maintain the quiescent state of hippocampal stem cells.

The Id (Inhibitor of differentiation/DNA binding) proteins are known inhibitors of bHLH transcription factors such as Ascl1 (*Imayoshi and Kageyama, 2014*; *Ling et al., 2014*). Id proteins contain a conserved HLH domain with which they dimerize with some bHLH proteins. However, they lack a

DNA binding domain and therefore prevent bHLHs with which they interact from binding DNA and other bHLH factors (*Benezra et al., 1990*). For instance, Id proteins have previously been shown to sequester E proteins, the dimerization partners of Ascl1. The resulting monomeric form of Ascl1 can no longer bind DNA and is furthermore rapidly degraded by the proteasome (*Shou et al., 1999*; *Viñals et al., 2004*). In mammals, the Id family comprises four genes, *Id1-4*. The *Id* genes are expressed in multiple tissues during development and in adult stem cell niches, and have been shown to promote stemness and proliferation in different systems, including in hematopoietic stem cells and in stem cells of the adult SVZ (*Niola et al., 2012*; *Singh et al., 2018*). However single cell transcriptome analysis has also shown that expression of Id3 and Id4 in particular, is highly enriched in quiescent hippocampal NSCs in vivo, thus linking Id genes with NSC quiescence (*Hochgerner et al., 2018*; *Shin et al., 2015*).

Here we show that Ascl1 mRNA is expressed by hippocampal stem cells independently of their proliferative or quiescent states, but that only active stem cells reach significant levels of Ascl1 protein. This non-transcriptional regulation of Ascl1 is recapitulated in hippocampal stem cell cultures in vitro, where the quiescence-inducing factor BMP4 has no effect on Ascl1 mRNA expression but is sufficient to reduce Ascl1 protein levels. We performed a gene expression screen in these cells and found that Id4 is strongly induced in quiescent NSCs. Accordingly, analysis of the expression of Id1-4 proteins in the hippocampus also showed that Id4 is expressed by the highest percentage of NSCs. We demonstrated that Id4 sequesters the Ascl1 heterodimerization partner E47 and that the resulting Ascl1 monomers are rapidly degraded by the proteasome. Therefore, Id4 blocks the pro-activation transcriptional program driven by Ascl1 and keeps stem cells quiescent. Indeed, elimination of Id4 from the adult brain results in increased Ascl1 protein levels in stem cells of the hippocampus and in their rapid entry into the cell cycle, and also leads to an increase in Id3 expression that might partially compensate for the loss of Id4 and suppresses Ascl1 protein in the absence of that factor.

## Results

### Ascl1 is transcribed in quiescent stem cells of the hippocampus

To investigate how *Ascl1* expression is regulated in hippocampal NSCs, we first assessed the transcriptional activity of the *Ascl1* locus using the *Ascl1^KIGFP* mouse reporter line, in which the GFP reporter replaces the *Ascl1* coding sequence and marks cells that transcribe the *Ascl1* gene (*Leung et al., 2007*). For clarity, hippocampal stem cells in vivo will be called hereafter radial glia-like cells (RGLs) while hippocampal stem cells in culture will be called NSCs. We identified RGLs by their expression of glial fibrillary acidic protein (GFAP), localization of their nucleus in the subgranular zone of the DG and presence of a radial process extending towards the molecular layer. We found that 82.3 ± 3.8% of all RGLs were positive for GFP in the hippocampus of P70 *Ascl1^KIGFP* mice and therefore transcribed *Ascl1*. In contrast, only 1.9 ± 0.3% of these cells expressed Ascl1 protein at a level detected with anti-Ascl1 antibodies (*Figure 1A and B*). Notably, 83.8 ± 4.1% of the RGLs that did not express Ki67 and were therefore quiescent expressed GFP, indicating a transcriptionally active *Ascl1* locus (*Figure 1C and D*). Moreover, GFP immunolabeling intensity was comparable in active Ki67+ RGLs and quiescent Ki67- RGLs (*Figure 1E*). We confirmed the presence of *Ascl1* transcripts at similar levels in quiescent and active RGLs using single molecule in situ hybridization (*Figure 1F,G*). These results show that, unexpectedly, *Ascl1* is already expressed in quiescent RGLs and that NSC activation is not accompanied by the induction or marked upregulation of *Ascl1* transcription. The finding that quiescent and proliferating RGLs transcribe the *Ascl1* gene at comparable levels but only proliferating RGLs express detectable levels of Ascl1 protein, indicates that Ascl1 protein expression in quiescent hippocampal RGLs is regulated by a non-transcriptional mechanism.

### Ascl1 is regulated post-translationally in quiescent NSC cultures

To investigate the mechanism regulating Ascl1 protein levels in quiescent hippocampal stem cells, we used an established cell culture model of NSC quiescence (*Martynoga et al., 2013*; *Mira et al., 2010*; *Sun et al., 2011*). The signalling molecule BMP4 has been shown to contribute to the maintenance of NSC quiescence in the hippocampus (*Bonaguidi et al., 2008*; *Mira et al., 2010*). BMP4 is also able to induce a reversible state of cell cycle arrest in embryonic stem cell-derived NSC cultures

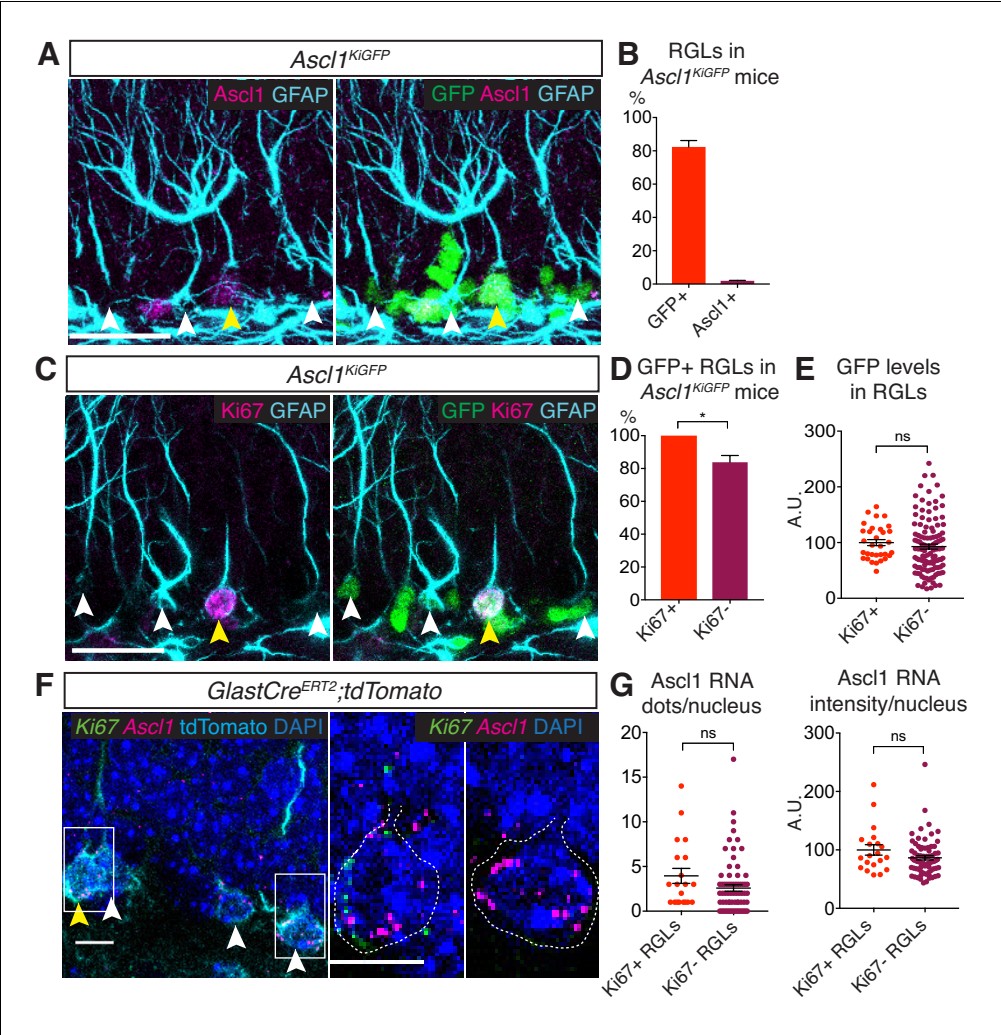

**Figure 1.** Ascl1 is transcribed in both quiescent and proliferating hippocampal stem cells. (**A**) Immunolabeling for GFP, Ascl1 and GFAP in the subgranular zone (SGZ) of the dentate gyrus (DG) of *Ascl1^{KIGFP}* reporter mice. White arrows indicate GFP+Ascl1- RGLs; yellow arrows indicate GFP+Ascl1+ RGLs. Scale bar, 30 μm. (**B**) Quantification of the data shown in (**A**). The widespread GFP expression indicates that Ascl1 is transcribed in most RGLs (radial GFAP+ cells) in *Ascl1^{KIGFP}* mice while Ascl1 protein is only detectable in a small fraction of RGLs. n = 3. (**C**) Immunolabeling for GFP, Ki67 and GFAP in the SGZ of the DG of *Ascl1^{KIGFP}* reporter mice. White arrows indicate GFP+Ki67- RGLs; yellow arrows indicate GFP+Ki67+ RGLs. Scale bar, 30 μm. (**D, E**) Quantification of the data in (**C**). Most quiescent (Ki67-) RGLs express GFP and therefore transcribe *Ascl1* (p=0.017) (**D**) and the levels of GFP are not significantly different in quiescent and proliferating RGLs (p=0.429) (**E**), indicating that Ascl1 is transcribed uniformly in the two RGL populations. n = 3. (**F**) RNA in situ hybridization by RNAscope with an Ascl1 probe (magenta) and a Ki67 probe (green) and immunolabeling for tdTomato to mark RGLs in the SGZ of the DG. To label RGLs with tdTomato, *Glast-Cre^{ERT2};tdTomato* mice were injected once at P60 with 4-hydroxytamoxifen, and analyzed 48 hr later. White arrows indicate RGLs positive for Ascl1 RNA staining; yellow arrows show RGLs positive for both Ascl1 and Ki67 RNA. Magnifications of the RGLs marked by white boxes are shown on the right, highlighting an RGL positive for both Ascl1 and Ki67 RNA, and an RGL positive for only Ascl1 RNA. Dotted lines show the outline of the tdTomato signal. Scale bar, 10 μm. n = 5. (**G**) Quantification of the data in (**F**). Ascl1 transcripts are found at a similar level in quiescent (Ki67-) and proliferating (Ki67+) RGLs (dots/nucleus p=0.101; intensity/nucleus p=0.112). Note the high variability in the levels of Ascl1 mRNA, which could be a reflection of the oscillatory nature of Ascl1 expression (*Imayoshi et al., 2013*). n = 5. Error bars represent mean ± SEM. Significance values: ns, p>0.05; *, p<0.05; **, p<0.01; ***, p<0.001; ****, p<0.0001.

DOI: https://doi.org/10.7554/eLife.48561.003

The following source data is available for figure 1:

*Figure 1 continued on next page*

*Figure 1 continued*

**Source data 1.** Original quantification of Ascl1kiGFP, Ascl1 antibody staining and Ascl1 RNA in active and quiescent RGLs.

DOI: https://doi.org/10.7554/eLife.48561.004

---

(*Martynoga et al., 2013*; *Mira et al., 2010*). Similarly, we found that NSCs originating from the adult hippocampus and maintained in culture in the presence of FGF2 stopped proliferating and entered a reversible quiescent state when exposed to BMP4 (*Figure 2—figure supplement 1A–D*). RNA sequencing analysis revealed that 1839 genes were differentially expressed between NSCs in proliferating and quiescent states (*Figure 2—figure supplement 1E,F*). Ascl1 RNA levels were not significantly different between these two conditions as verified by QPCR (*Figure 2A*). The intensity of GFP in cultured hippocampal NSCs derived from *Ascl1$^{KIGFP}$* mice was also comparable in proliferating and quiescent conditions (*Figure 2B,C*), as for GFP expression in the hippocampus of *Ascl1-$^{KIGFP}$* mice (*Figure 1E*). In contrast, Ascl1 protein levels were strongly reduced in BMP-treated quiescent NSCs (*Figure 2D–F*), resembling the absence of Ascl1 protein in quiescent hippocampal RGLs in vivo (*Figure 1B*). Treatment of quiescent NSCs with proteasome inhibitors significantly increased the levels of Ascl1 protein, suggesting that Ascl1 mRNA is translated in quiescent NSCs but Ascl1 protein is rapidly degraded in a proteasome-dependent manner (*Figure 2G* and *Figure 2—figure supplement 1G*). We previously showed that Ascl1 protein is targeted for proteasomal degradation by the E3 ubiquitin ligase Huwe1 in proliferating hippocampal NSCs (*Urbán et al., 2016*). We therefore asked whether Huwe1 is also responsible for the degradation of Ascl1 in quiescent NSCs. We found that Ascl1 protein was similarly reduced in *Huwe1* mutant and control NSC cultures upon addition of BMP4 (*Figure 2—figure supplement 1H,I*), demonstrating that a *Huwe1*-independent mechanism prompts the down-regulation of Ascl1 protein in quiescent cultured NSCs. Together, these results establish BMP-treated NSC cultures as an appropriate model to characterize the mechanisms controlling Ascl1 protein levels in quiescent hippocampal stem cells.

## Id4 is highly expressed in quiescent hippocampal stem cells in culture and in vivo

We screened our RNA-Seq data for potential Ascl1 inhibitory factors induced in quiescent conditions (*Figure 3—figure supplement 1*). The four *Id* genes *Id1-4* were strongly induced in quiescent NSC cultures (*Figure 3—figure supplement 1A*). Since Id proteins sequester E proteins, resulting in the degradation of Ascl1 monomers, they are strong candidates to regulate Ascl1 post-translationally in hippocampal stem cells (*Shou et al., 1999*; *Viñals et al., 2004*). Although the transcripts for the four *Id* genes were induced by BMP4 in NSC cultures (*Figure 3A*), Id2 and Id3 were already expressed at high levels in proliferating conditions and only Id1 and Id4 were clearly upregulated at the protein level upon addition of BMP (*Figure 3B–E* and *Figure 3—figure supplement 1B,C*). Of those, Id4 expression was highly variable and was absent from cells presenting high levels of Ascl1, suggesting a possible negative regulatory relationship between the two proteins, while Id1 was also expressed at various levels but did not anti-correlate with Ascl1 expression levels (*Figure 3C–F* and *Figure 3—figure supplement 1D–I*). Id4 therefore represented the most promising Id protein for dynamically regulating Ascl1 protein levels in BMP4-induced quiescent NSCs. Co-immunoprecipitation (co-IP) experiments in NSCs showed that the E protein E47 (a product of the *Tcf3/E2A* gene) interacts with Ascl1 in proliferating conditions but with Id4 in quiescence conditions (*Figure 3—figure supplement 1J*). However, Ascl1 protein levels were lower in quiescent cultures, making the co-IP results difficult to interpret. We therefore also carried an in vitro competition-binding assay with separately transfected E47, Ascl1 and Id4 gene products (*Figure 3G*), which confirmed that the interaction between Ascl1 and E47 is disrupted by the presence of Id4, whilst no interaction was detected between Id4 and Ascl1 (*Figure 3H*). Therefore, Id4 sequesters E proteins away from their binding partner Ascl1. In agreement with monomeric Ascl1 being more unstable than the heterodimer with E47, we found that the half-life of Ascl1 was reduced from 194 min in proliferating cells to 31 min in quiescent cells (*Figure 3—figure supplement 1K,L*).

Next, we examined the expression of Id proteins in the adult DG by immunohistochemistry. Id1, Id3 and Id4 proteins are clearly expressed in the SGZ of the DG where hippocampal stem cells are

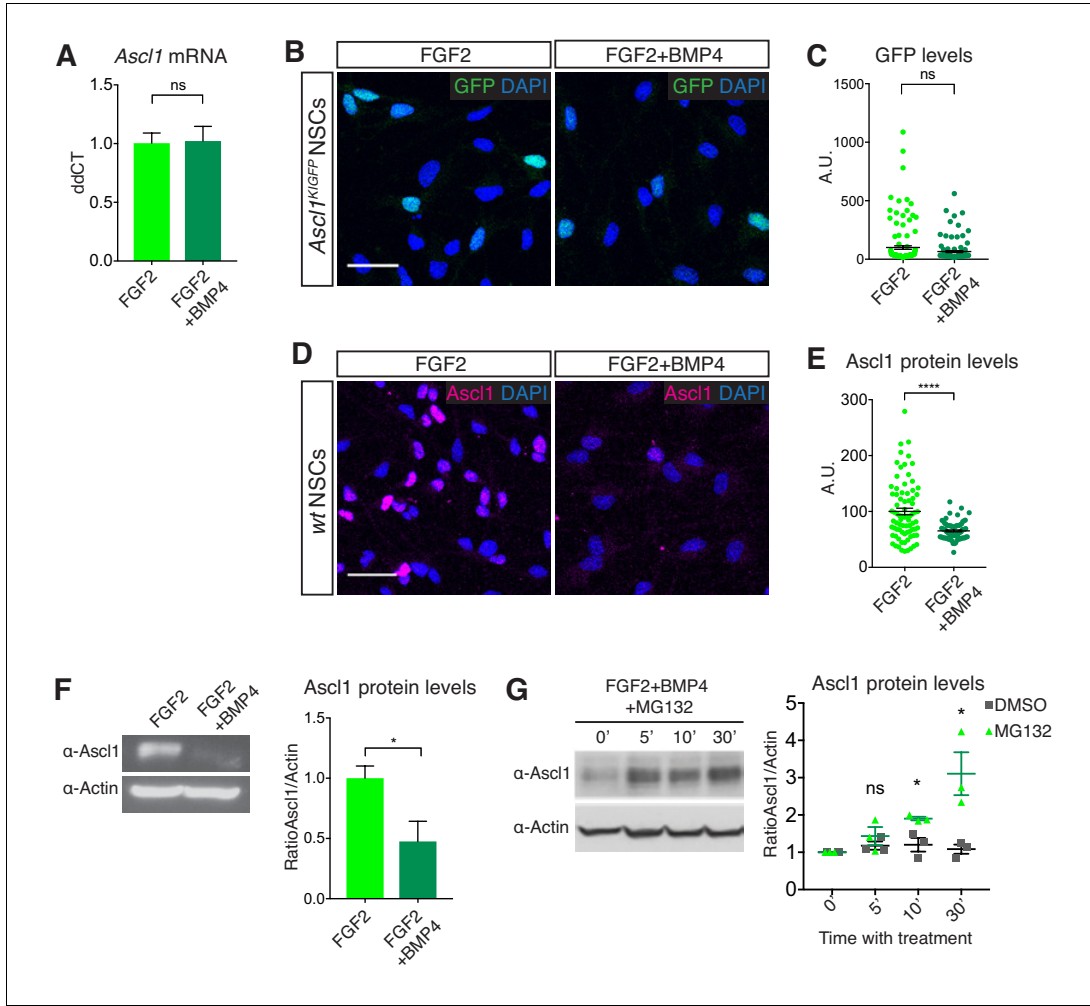

**Figure 2.** Ascl1 is regulated post-translationally in quiescent NSC cultures. (**A**) Transcript levels for *Ascl1* in DG-derived NSC cultures treated with FGF2 alone (proliferating NSCs) or FGF2 and BMP4 (quiescent NSCs) analyzed by QPCR. Ascl1 mRNA levels are unchanged in FGF2+BMP4-treated, quiescent NSCs (p=0.908). n = 3. (**B**) Immunolabeling for GFP and DAPI staining in FGF2 and FGF2+BMP4-treated NSC cultures originating from *Ascl1^KIGFP* mice. Scale bar, 30 μm. (**C**) Quantification of the data in (**B**). GFP, which reports transcription of the Ascl1 gene, is expressed at similar levels in proliferating and quiescent NSCs (p=0.058). The data show one representative experiment of n = 3. (**D**) Immunolabeling for Ascl1 and DAPI staining in FGF2- and FGF2+BMP4-treated NSC cultures. Scale bar, 30 μm. (**E**) Quantification of the data in (**D**). Ascl1 levels are high in many proliferating NSCs and not detectable in most quiescent NSCs (p=7.09E-8). The heterogeneity of Ascl1 expression in proliferating NSCs most likely reflects its oscillatory behaviour. n = 3. (**F**) Western blot analysis and quantification of Ascl1 in FGF2-treated and FGF2+BMP4-treated NSCs. BMP4 suppresses Ascl1 protein expression (p=0.0363). n = 3. (**G**) Western blot analysis and quantification of Ascl1 in FGF2+BMP4-treated NSCs after treatment with the proteasome inhibitor MG132 for different durations or with DMSO vehicle as a control. Ascl1 can be detected after proteasome inhibition in quiescent NSCs and is significantly increased compared to DMSO conditions from 10mins after MG132 treatment (5' p=0.387; 10' p=0.020; 30' p=0.026). n = 3. Significance calculated using Multiple t test Error bars represent mean ± SEM. Significance values: ns, p>0.05; *, p<0.05; **, p<0.01; ***, p<0.001; ****, p<0.0001. See also *Figure 2—figure supplement 1*.

DOI: https://doi.org/10.7554/eLife.48561.005

The following source data and figure supplement are available for figure 2:

**Source data 1.** Original quantification of Ascl1 mRNA and protein in proliferating and quiescent NSCs.
DOI: https://doi.org/10.7554/eLife.48561.007
**Figure supplement 1.** BMP4 induces reversible quiescence of adult hippocampal NSCs.
DOI: https://doi.org/10.7554/eLife.48561.006

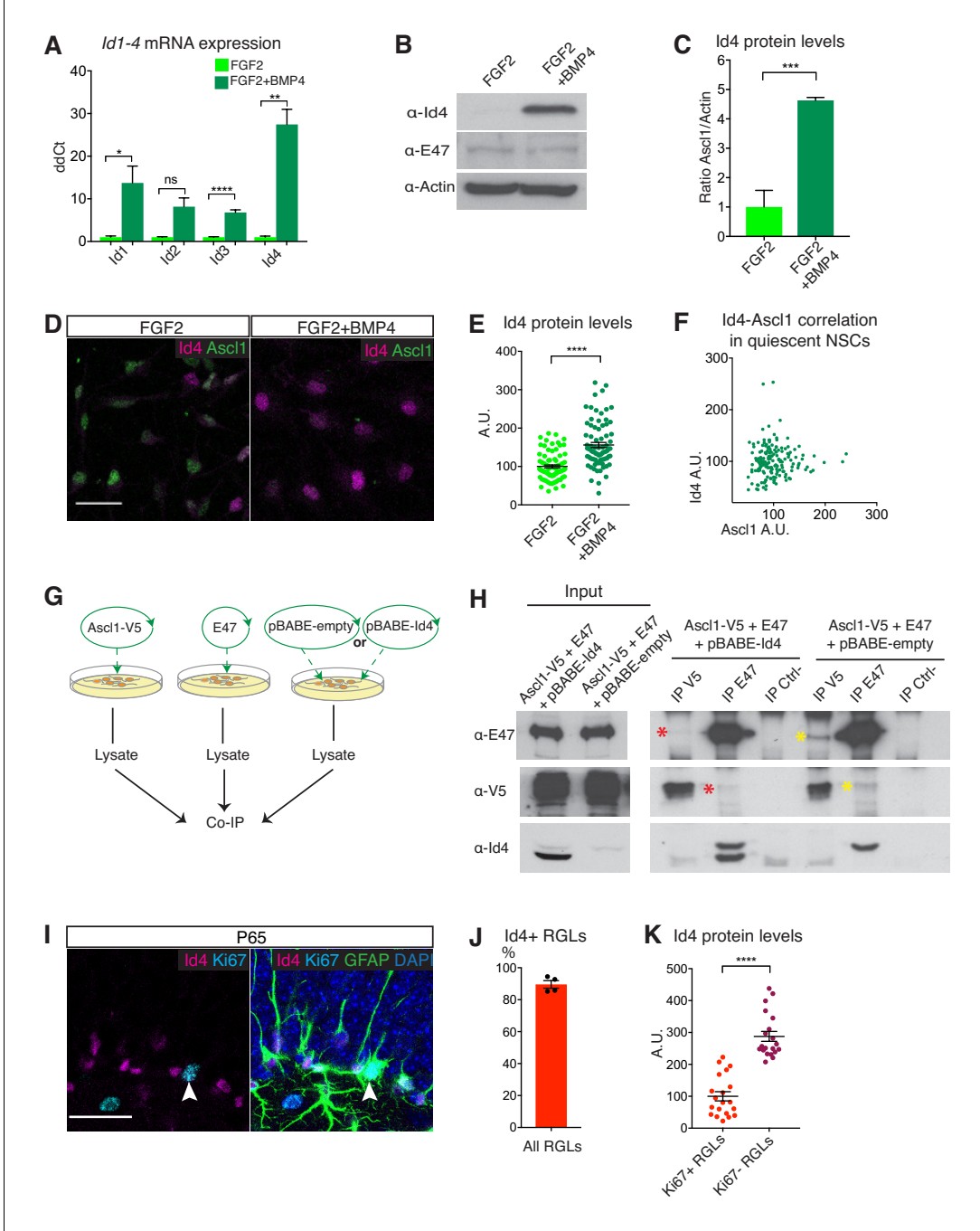

**Figure 3.** Id4 is a candidate regulator of Ascl1 protein expression in quiescent hippocampal stem cells. (**A**) Transcript levels for the four Id genes (*Id1, Id2, Id3, Id4*) in FGF2-treated and FGF2+BMP4-treated NSCs cultures analyzed by QPCR. BMP strongly induces the *Id* genes (*Id1* p=0.032; *Id2* p=0.074; *Id3* p=7.29E-4; *Id4* p=0.001). n = 3. (**B**) Western blot analysis of Id4 and E47 in FGF2-treated and FGF2+BMP4-treated NSCs. BMP4 upregulates Id4 protein expression; E47 expression is unchanged. n = 3. (**C**) Quantification of Id4 protein levels shown in (**B**) (p=7.53E-4). n = 3. (**D**) Immunolabeling for Id4 and Ascl1 in FGF2-treated and FGF2+BMP4-treated NSCs. Scale bar, 30 μm. (**E–F**) Quantifications of the data in (**D**). (**E**) BMP4 treatment increases Id4 protein levels in NSCs, detected by immunofluorescence (p=2.32E-11). n = 3. (**F**) Id4 protein is expressed at high levels in NSCs expressing low levels of Ascl1 protein. n = 3. (**G**) Scheme for the in vitro competition-binding assay in HEK293T cells between overexpressed Ascl1, E47 and Id4 or its empty vector. Cells were independently transduced and lysates mixed prior to co-immunoprecipitation (co-IP). (**H**) Western blot analysis of the in vitro competition-binding assay. When Id4 is not present in the lysate, co-IP between Ascl1 and E47 is detected (yellow asterisks). The addition of excess Id4 disrupts in vitro binding of Ascl1 to E47 (red asterisks). Inputs show overexpression of Ascl1, E47 and Id4. (**I**) Immunolabeling for Id4, Ki67 and GFAP and staining for DAPI in hippocampal RGLs. White arrow indicates an Id4+Ascl1+ RGL. Scale bar, 30 μm. (**J–K**) Quantification of the data in (**I**). Id4 is expressed in the majority of RGLs (**J**), and at high levels in quiescent (Ki67-) RGLs and low levels or is not expressed in proliferating (Ki67+) RGLs (**K**)

*Figure 3 continued on next page*

*Figure 3 continued*

(p=8.4E-11). n = 3. Error bars represent mean ± SEM. Significance values: ns, p>0.05; *, p<0.05; **, p<0.01; ***, p<0.001; ****, p<0.0001. See also **Figure 3—figure supplements 1** and **2**.

DOI: https://doi.org/10.7554/eLife.48561.008

The following source data and figure supplements are available for figure 3:

**Source data 1.** Original quantification of Id4 protein levels in proliferating and quiescent hippocampal NSCs and RGLs.

DOI: https://doi.org/10.7554/eLife.48561.011

**Figure supplement 1.** Dynamics of Id1-4 and Ascl1 expression in proliferating and quiescent NSCs.

DOI: https://doi.org/10.7554/eLife.48561.009

**Figure supplement 2.** Expression of Id1-4 and Tcf4 in RGLs in the dentate gyrus.

DOI: https://doi.org/10.7554/eLife.48561.010

located, while Id2 is enriched in granule neurons but not detected in the SGZ (*Figure 3I* and *Figure 3—figure supplement 2A–F*). Id1 is expressed by a substantial fraction of RGLs (47.5 ± 7.3%) and is enriched in proliferating cells (*Figure 3—figure supplement 2A–C*). Id3 is expressed by a small fraction of mostly quiescent RGLs (16.9 ± 1.9%), at similar levels in proliferating and quiescent RGLs (*Figure 3—figure supplement 2E–G*). In contrast, Id4 is expressed by the vast majority of RGLs (89.6 ± 2.3%), at high levels in quiescent hippocampal RGLs and at much lower levels in proliferating RGLs (*Figure 3I–K*), and co-localizes with *Ascl1* mRNA-expressing cells (*Figure 3—figure supplement 2H,I*). Expression of the three genes encoding Eeins (*Tcf3/E2A*, *Tcf4/Itf2/E2.2* and *Tcf12/Heb*) has been reported in RGLs in single cell RNA sequencing studies and we confirmed the presence of Tcf4/Itf2/E2.2 protein in RGLs in the DG (*Figure 3—figure supplement 2J,K*) (*Hochgerner et al., 2018*). Altogether, Id4 is a good candidate to suppress Ascl1 protein and promote quiescence in hippocampal stem cells via Eein sequestration, both in culture and in vivo.

## Id4 promotes the degradation of Ascl1 protein and induces a quiescence-like state in NSCs

To address the role of Id4 in Ascl1 regulation and in hippocampal stem cell quiescence, we first asked whether forcing the expression of Id4 in proliferating NSCs would be sufficient to reduce Ascl1 protein level and induce a quiescent state (*Figure 4A*). Id4 expression was low in control proliferating NSCs (*Figure 3B–D*) and was strongly increased after transfection with an Id4 expression construct (*Figure 4C*). Id4-transfected NSCs maintained Ascl1 mRNA at levels similar to those of control NSCs but showed markedly reduced Ascl1 protein levels (*Figure 4C–E*). Moreover, transfection of Id4 resulted in a significant decrease in NSC proliferation (*Figure 4F–I* and *Figure 4—figure supplement 1A, B*). This decrease was not due to differentiation, since Id4-expressing cells retained expression of the stem cell markers Sox2 and Nestin (*Figure 4—figure supplement 1C,D*). The effects of Id4 protein on Ascl1 expression and NSC proliferation suggest that induction of Id4 in BMP-treated NSCs contributes to the degradation of Ascl1 protein and the induction of quiescence (*Figure 4A*). We then asked whether over-expression of Id1-3 beyond their endogenous levels in proliferating NSCs could also reduce Ascl1 protein levels, by transfecting NSCs in parallel with expression construct for each of the Ids. Over-expressing Id1, Id2 or Id3 also suppressed Ascl1 protein levels, although Id4 overexpression was most effective (*Figure 4—figure supplement 1E*).

Next, we asked whether inactivating Id proteins could stabilize Ascl1 protein and revert some aspects of the quiescent state in BMP-treated NSCs. Knockdown of Id1-4 in quiescent NSCs by transfection of siRNAs targeting each Id gene separately or by co-transfection of Id4-siRNA with either Id1-, Id2- or Id3-siRNA, did not significantly affect Ascl1 protein levels (*Figure 4—figure supplement 1F*) despite a significant knockdown of each gene at both mRNA and protein levels (*Figure 4—figure supplement 1G–H*). It is worth noting that Id4 knockdown resulted in an increase in the protein levels of Id1 and Id3 (*Figure 4—figure supplement 1G*) without affecting their mRNA expression (*Figure 4—figure supplement 1H*), suggesting that Id4 may suppress Id1 and Id3 protein expression. Since Id1-3 can suppress Ascl1 protein when overexpressed (*Figure 4—figure supplement 1E*), the upregulation of Id1 and Id3 in Id4-silenced cells might constitute a compensatory mechanism that maintains Ascl1 protein at low levels. Because NSCs express the four Id proteins, which have redundant functions, we next chose to neutralise all of them by overexpressing the E protein E47. Since Id proteins have been shown to strongly bind E proteins, we reasoned that an

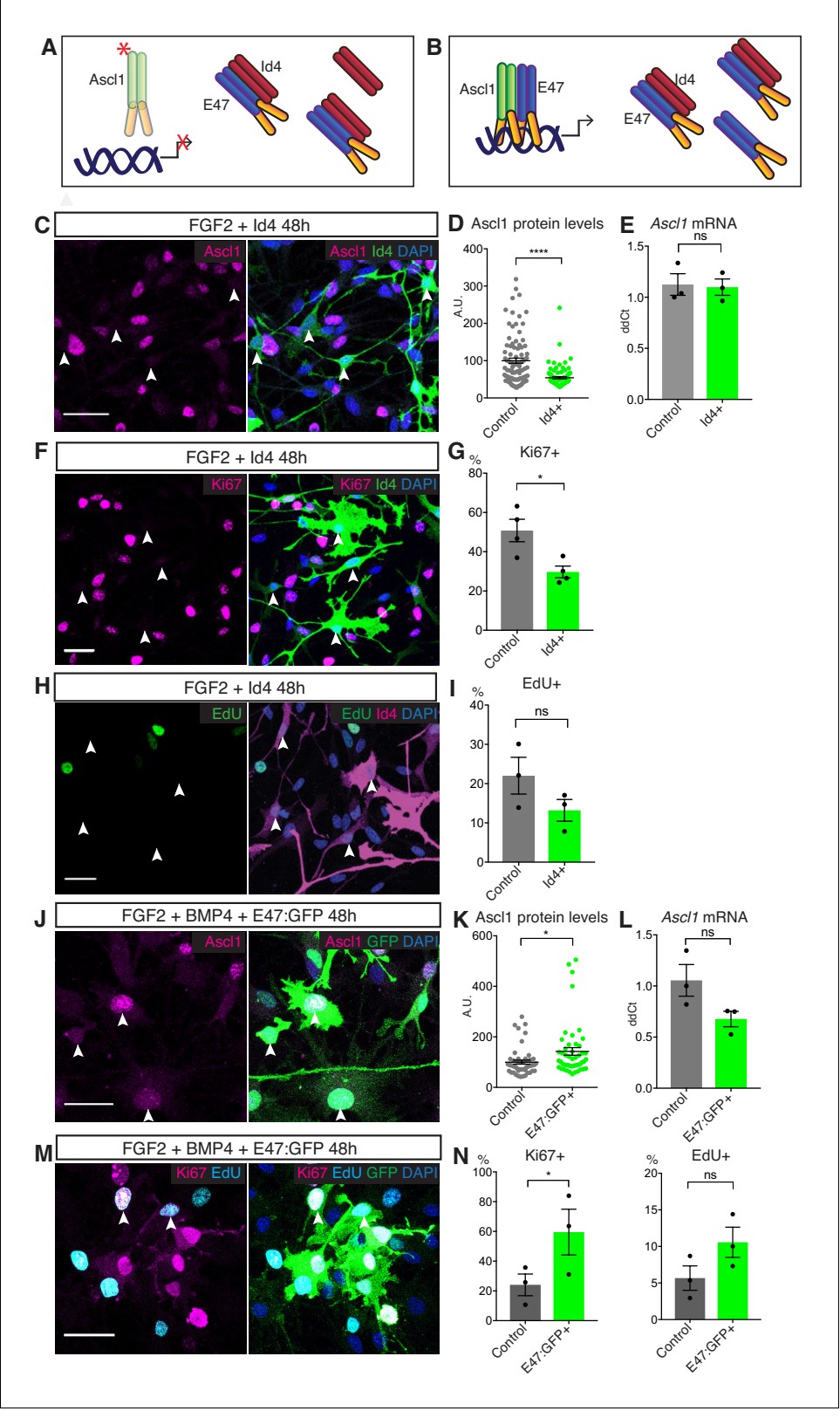

**Figure 4.** Id4 suppresses Ascl1 expression and cell proliferation in NSC cultures. (**A**) Model of Ascl1 monomerization and elimination following Id4 overexpression in proliferating NSC cultures. (**B**) Model of Id protein titration by E47 overexpression in quiescent NSC cultures. (**C**) Immunolabeling for Ascl1 and Id4 and staining for DAPI in Id4-overexpressing, FGF2-treated NSCs. White arrows show low Ascl1 levels in Id4-overexpressing cells.
*Figure 4 continued on next page*

*Figure 4 continued*

Scale bar, 30 μm. (D) Quantification of the data in (C). Ascl1 protein expression is strongly reduced by Id4 overexpression. The data show one representative experiment, n = 3 (p=3.05E-9). (E) *Ascl1* mRNA levels in FACS sorted FGF2-treated NSCs transfected with a GFP-expressing control or Id4-expression construct. *Ascl1* mRNA levels are not changed by Id4 overexpression (p=0.873). n = 3. (F) Immunolabeling for Ki67 and Id4 in Id4-overexpressing, FGF2-treated NSCs. White arrows indicate absence of Ki67 in Id4-overexpressing cells. Scale bar, 30 μm. (G) Quantification of the data in (C). Id4 overexpression reduces NSC proliferation (p=0.050). n = 3. (H) Staining for EdU and immunolabeling for Id4 in Id4-overexpressing, FGF2-treated NSCs. EdU was administered to the cultured cells one hour before fixation. White arrows indicate absence of EdU in Id4-overexpressing cells. Scale bar, 30 μm. (I) Quantification of the data in (H). Id4 overexpression reduces the fraction of NSCs in S-phase (p=0.114). n = 3. (J) Immunolabeling for Ascl1 and GFP with DAPI staining, in E47:GFP-overexpressing, FGF2+BMP4-treated NSCs. White arrows indicate Ascl1-positive, E47-overexpressing quiescent cells. Scale bar, 30 μm. (K) Quantification of the data in (J) (p=0.013). (L) Titration of Id proteins by E47 results in a significant increase in Ascl1 protein expression without significant change in Ascl1 RNA levels (p=0.075). n = 3 (M) Immunolabeling for GFP and Ki67 and staining for EdU and DAPI in E47:GFP-overexpressing, FGF2+BMP4-treated NSCs. White arrows indicate E47-overexpressing quiescent cells positive for Ki67 and EdU. Scale bar, 30 μm. (N) Quantification of the data in (M). Titration of Id proteins by E47 reverts the proliferation arrest of BMP4-treated NSCs (%Ki67+ p=0.048; %EdU+ p=0.085). n = 3. Error bars represent mean ± SEM. Significance values: ns, p>0.05; *, p<0.05; **, p<0.01; ***, p<0.001; ****, p<0.0001. See also *Figure 4—figure supplement 1*.
DOI: https://doi.org/10.7554/eLife.48561.012

The following source data and figure supplement are available for figure 4:

**Source data 1.** Original quantification of Ascl1, Ki67 and EdU in Id4- and E47-overexpressing NSCs.
DOI: https://doi.org/10.7554/eLife.48561.014
**Figure supplement 1.** Analysis of Id1-4 overexpression and siRNA knockdown in NSCs.
DOI: https://doi.org/10.7554/eLife.48561.013

excess amount of E47 should sequester Id proteins into E47-Id complexes, allowing the formation of Ascl1-E47 complexes and the stabilization of Ascl1 (*Figure 4B*). Indeed, overexpression of E47 in BMP-treated NSCs resulted in an increase in the levels of Ascl1 protein without significantly affecting Ascl1 mRNA levels (*Figure 4J–L*). Overexpression of E47 was also sufficient to partially revert the cell cycle arrest of BMP-treated NSCs, and we observed a strong correlation between Ascl1 protein levels and the proliferative state of the cells (*Figure 4M,N* and *Figure 4—figure supplement 1I,J*). Together, these results support a model whereby induction of high levels of Id proteins by BMP4 in quiescent NSCs promotes the degradation of Ascl1 by sequestering its dimerization partners (*Figure 4B*). They also raise the possibility that suppression of the transcriptional activity of Ascl1 is a key feature of the induction of quiescence by Id proteins.

## Quiescence is characterized by a downregulation of Ascl1 target genes

All four Id proteins can reduce Ascl1 protein expression when overexpressed in NSCs, but the mutually exclusive expression of Id4 and Ascl1 proteins suggested that Id4 may have the most important role among endogenous Id proteins for the regulation of Ascl1 in quiescent NSCs (*Figure 3F* and *Figure 3—figure supplement 1D–I*). To investigate the mechanism by which Id4 induces quiescence in NSCs, we compared the transcriptome of Id4-overexpressing and control proliferating NSCs using RNA-Seq. Expression of Id4 resulted in the up-regulation of 806 genes and down-regulation of 823 genes (*Figure 5A*). Expression of *Ascl1*, *Tcf3*, *Tcf4*, *Tcf12*, *Hes1*, *Hes5* and *Hey1* were not significantly changed by *Id4* overexpression in our data set (*Figure 5—figure supplement 1A*). Id4-regulated genes represented 44.2% of the genes regulated by BMP4 in NSCs, including 31.1% of the upregulated and 56.2% of the downregulated genes, indicating that Id4 has an important role in the induction of the gene expression program of quiescence downstream of BMP4 (*Figure 5B* and *Figure 5—figure supplement 1B–D*). The genes commonly regulated by Id4 and BMP4 are involved in cell cycle (downregulated) and cell adhesion (upregulated) (*Figure 5C–D*), which are hallmarks of the NSC quiescent state (*Llorens-Bobadilla et al., 2015*; *Martynoga et al., 2013*; *Shin et al., 2015*). Direct transcriptional targets of Ascl1 were strongly downregulated in Id4-overexpressing NSCs, including genes with important roles in cell cycle progression such as *Skp2*, *Cdk1*, *Cdk2* and *Foxm1*, as well as other canonical Ascl1 targets such as *Dll1* and *Dll3* (*Castro et al., 2011*; *Martynoga et al., 2013*) (*Figure 5E*). Overall, our analysis indicates that induction of Id4 by BMP4 and the subsequent

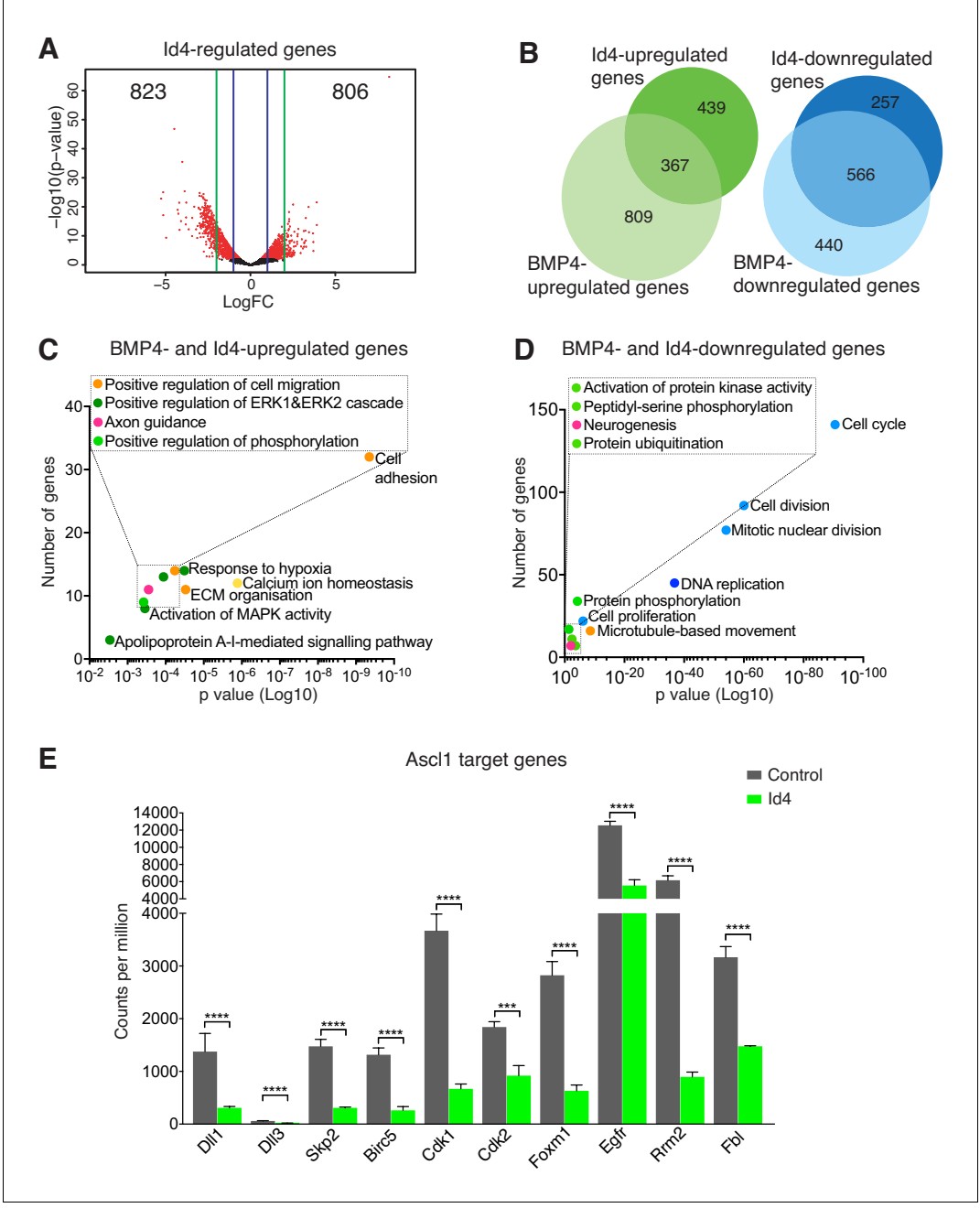

**Figure 5.** Id4 regulation of Ascl1 targets contributes to BMP-induced NSC quiescence. (**A**) Volcano plot displaying gene expression changes between control and Id4-overexpressing FGF2-treated NSCs analyzed by RNA-Seq. (**B**) Venn diagrams indicating the number of genes up- and down-regulated by addition of BMP4 or Id4-overexpression or both in FGF2-treated NSCs. (**C–D**) Gene Ontology terms associated with genes up- or down-regulated by both addition of BMP4 and Id4-overexpression in FGF2-treated cultures. Dots are colored based on their ontology terms; light blue: cell cycle/division; dark blue: DNA repair/replication; light green: Protein phosphorylation/modification; dark green: signalling, transcription; orange: adhesion/cytoskeleton; yellow: ion-related; pink: brain/nervous system related. (**E**) Downregulation of Ascl1 target genes in FGF2-treated cultures overexpressing Id4 and analyzed by RNA-Seq, including canonical Ascl1 targets (*Dll1* and *Dll3*), genes involved in cell cycle regulation (*Skp2, Cdk1, Cdk2* and *Foxm1*), RGL activation (*Egfr*) and other Ascl1 targets previously identified in NSCs (*Birc5, Rrm2* and *Fbl*). ($p$ values in order of genes: 3.04E-11, 2.31E-6, 9.49E-18, 1.32E-12, 8.47E-19, 3.62E-4, 6.72E-14, 6.25E-6, 1.13E-23, 4.96E-6). n = 3. Error bars represent mean ± SEM. Significance values: ns, $p > 0.05$; *, $p < 0.05$; **, $p < 0.01$; ***, $p < 0.001$; ****, $p < 0.0001$. See also *Figure 5—figure supplement 1*.

DOI: https://doi.org/10.7554/eLife.48561.015

*Figure 5 continued on next page*

*Figure 5 continued*

The following source data and figure supplement are available for figure 5:

**Source data 1.** Significantly regulated genes and gene ontology analysis, and CPM values for Ascl1 target genes in Id4-overexpressing NSCs.

DOI: https://doi.org/10.7554/eLife.48561.017

**Figure supplement 1.** RNAseq analysis of proliferating NSCs overexpressing Id4.

DOI: https://doi.org/10.7554/eLife.48561.016

degradation of Ascl1 results in the downregulation of its targets, leading to the cell cycle arrest of NSCs.

## Loss of Id4 in vivo activates quiescent adult hippocampal RGLs

In light of the role of Id4 in inducing a quiescent-like state in NSCs, and since Id4 is highly expressed in stem cells in the adult hippocampus and is particularly enriched in quiescent RGLs, we next assessed the role of *Id4* in the maintenance of the quiescent state of RGLs in vivo by analysing the hippocampus of mice carrying a conditional mutant allele of *Id4* (*Id4$^{fl}$*) (*Best et al., 2014*) (*Figure 6A* and *Figure 6—figure supplement 1A*). To eliminate *Id4* from RGLs, we crossed *Id4$^{fl/fl}$* mice with the Glast-CreERT2 deleter line (*Mori et al., 2006*) and the tdTomato reporter line (*Madisen et al., 2010*) (*Figure 6A*). We administered tamoxifen to the triple transgenic mice for 5 days, which resulted in complete elimination of Id4 protein (*Figure 6—figure supplement 1B*) and we analyzed the brains immediately after (*Id4$^{cKO}$* mice; *Figure 6A*). The fraction of RGLs expressing Ascl1 increased from 4.4 ± 0.5 in control mice to 15.3 ± 2.7 in *Id4$^{cKO}$* mice, with heterozygote mice showing only a small and non-significant increase in the fraction of Ascl1$^+$ RGLs (*Figure 6B,C*). Ascl1 protein levels were also upregulated in Ascl1-expressing RGLs from *Id4$^{cKO}$* mice while mRNA levels, measured by single molecule in situ hybridization, were lower in RGLs from *Id4$^{cKO}$* mice than in control mice (*Figure 6D* and *Figure 6—figure supplement 1C*). The fraction of proliferating RGLs increased from 4.1 ± 0.5 in control mice to 13.7 ± 2.0 in *Id4$^{cKO}$* mice, while heterozygote mice were indistinguishable from control mice (*Figure 6F,G*). Ascl1 expression was strongly correlated with Ki67 expression in RGLs in control and, particularly, in *Id4$^{cKO}$* mice, supporting the direct link between Ascl1 upregulation and RGL activation (*Figure 6—figure supplement 1D–F* and *Andersen et al., 2014*; *Urbán et al., 2016*).

When *Id4$^{cKO}$* mice were analyzed 30 days after tamoxifen administration and *Id4* deletion, the rate of proliferation of RGLs remained significantly higher than in control mice, although the difference was smaller (3-fold at 5 days and two-fold at 30 days; *Figure 6H*). Similarly, the fraction of Ascl1+ RGLs at 30 days post-*Id4* deletion, was increased to a lesser extent, and non-significantly, than at 5 days (*Figure 6E*). Since RGL activation is linked to the depletion of the RGL pool (*Encinas et al., 2011*; *Pilz et al., 2018*), we quantified the total number of RGLs in *Id4$^{cKO}$* and control mice 30 days after *Id4* deletion and found no difference between genotypes (*Figure 6—figure supplement 1G*). To determine whether deletion of *Id4* might trigger compensatory mechanisms, we examined the expression of the other Id proteins in *Id4$^{cKO}$* mice and found that Id1 and particularly Id3 were strongly upregulated in RGLs in these mice at both 5 days and 30 days after *Id4* deletion (*Figure 6I–L* and *Figure 6—figure supplement 1H,I*). We also analyzed the co-expression of Id3 protein, *Id4* mRNA and *Ascl1* mRNA by in situ and immunostaining in wild-type mice to determine whether Id3 could suppress Ascl1 protein independently of Id4 in a subset of RGLs that co-express Id3 and *Ascl1* and are negative for Id4. We found that the majority of Id3+GFAP+ RGLs co-express *Ascl1* mRNA (*Figure 6—figure supplement 1J,L*) and of the Id3+Ascl1+ cells, the vast majority also express *Id4* (*Figure 6—figure supplement 1K,M*). This suggests that Id3 might only regulate Ascl1 protein independently of Id4 in a very small number of RGLs, but may become functionally relevant and compensate for loss of Id4 in *Id4$^{cKO}$* mice. Together, these findings demonstrate that Id4 expression in hippocampal RGLs contributes to the suppression of Ascl1 protein expression and the maintenance of quiescence, and suggest that compensatory mechanisms involving the upregulation of other Id proteins maintain partially RGL quiescence in the absence of Id4.

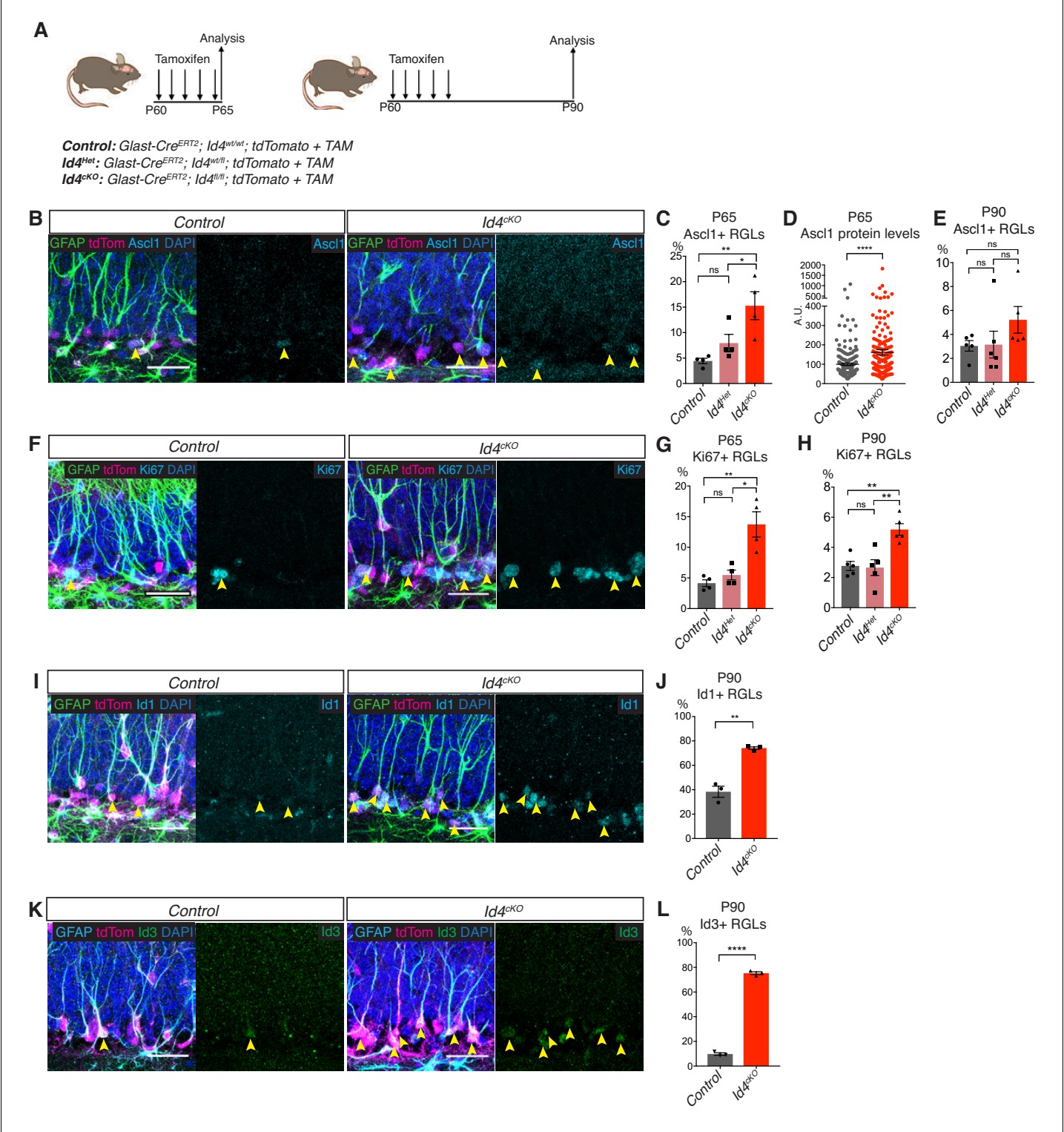

**Figure 6.** Loss of Id4 results in activation of quiescent RGLs in the adult hippocampus. (A) Design of the experiment for acute and long-term deletion of Id4 from RGLs of the adult hippocampus using *Id4^cKO* mice. (B) Immunolabeling for GFAP, tdTomato, Ascl1 and DAPI staining in control and *Id4^cKO* mice after 5 days of tamoxifen administration. Yellow arrows indicate Ascl1-positive RGLs. Scale bar, 30 μm. (C–D) Quantification of Ascl1 protein in tdTomato+ RGLs in control, *Id4^Het* and *Id4^cKO* mice after 5 days of tamoxifen administration. Loss of both copies of Id4 results in increases in the number of Ascl1-expressing cells and in the levels of Ascl1 protein in RGLs (*Control vs Het p*=0.3276; *Control vs cKO p*=0.0067; *Het vs cKO p*=0.0381; protein levels p=2.01E-5). n = 4 for control, *Id4^Het* and *Id4^cKO* mice. (E) Quantification of Ascl1 protein in tdTomato+ RGLs control, *Id4^Het* and *Id4^cKO*

*Figure 6 continued on next page*

*Figure 6 continued*

mice 30 days after tamoxifen administration. The percentage of RGLs positive for Ascl1 is increased in *Id4^cKO* mice compared with control mice 30 days after Id4 deletion (*Control vs Het* p=0.996; *Control vs cKO* p=0.311; *Het vs cKO* p=0.315). n = 5 for control and *Id4^Het* mice, n = 6 for *Id4^cKO* mice. **(F)** Immunolabeling for GFAP, tdTomato, Ki67 and DAPI staining in control and *Id4^cKO* and control mice after 5 days of tamoxifen administration. Yellow arrows indicate Ki67-positive RGLs. Scale bar, 30 μm. **(G–H)** Quantification of the fraction of Ki67+ tdTomato+ RGLs in control, *Id4^Het* and *Id4^cKO* mice, 5 days **(G)** and 30 days **(H)** following tamoxifen administration. The percentage of Ki67+ tdTomato+ RGLs is strongly increased following acute deletion of both copies of the Id4 allele, and remained significantly increased, albeit to a lesser extent, following long-term deletion. (*Control vs Het* P65 p=0.7595, P90 p=0.980; *Control vs cKO* P65 p=0.0049, P90 p=0.0036; *Het vs cKO* p=0.0101, P90 p=0.0026). n = 4 for P65 control, *Id4^Het* and *Id4^cKO* mice at P65; n = 5 for P90 control, *Id4^Het* and *Id4^cKO* mice. **(I)** Immunolabeling for GFAP, tdTomato, Id1 and DAPI staining in control and *Id4^cKO* and control mice 30 days after tamoxifen administration. Yellow arrows indicate Id1-positive RGLs. Scale bar, 30 μm. **(J)** Quantification of the fraction of Id1+ tdTomato+ RGLs 30 days after tamoxifen administration in control and *Id4^cKO* mice. Loss of Id4 results in a 2-fold increase in the fraction of tdTomato+ RGLs positive for Id1 immunoreactivity, from 38.3 ± 4.5% to 74.1 ± 1.0% (p=0.0016). n = 3 for both control and *Id4^cKO*. **(K)** Immunolabeling for GFAP, tdTomato, Id3 and DAPI staining in control and *Id4^cKO* and control mice 30 days after tamoxifen administration. Yellow arrows indicate Id3-positive RGLs. Scale bar, 30 μm. **(L)** Quantification of the fraction of Id3+ tdTomato+ RGLs in **(K)**. Id3 is increased by more than 8-fold in tdTomato+ RGLs following Id4 deletion, from 9.7 ± 1.0% to 75.3 ± 1.1% (p=1.87E-6). n = 4 for control mice and n = 3 for *Id4^cKO* mice. Error bars represent mean ± SEM. Significance values: ns, p>0.05; *, p<0.05; **, p<0.01; ***, p<0.001; ****, p<0.0001. See also *Figure 6—figure supplement 1*.

DOI: https://doi.org/10.7554/eLife.48561.018

The following source data and figure supplement are available for figure 6:

**Source data 1.** Original quantification of Ascl1, Ki67, Id1 and Id3 protein levels in RGLs following Id4 deletion.
DOI: https://doi.org/10.7554/eLife.48561.020

**Figure supplement 1.** Expression of Id4, Ascl1, Ki67, Id1 and Id3 in RGLs following loss of Id4.
DOI: https://doi.org/10.7554/eLife.48561.019

## Discussion

In this study, we show that the repressor protein Id4 promotes the maintenance of adult hippocampal stem cells (RGLs) in a quiescent state. The function of Id4 in maintenance of RGL quiescence is in remarkable contrast with its role in promoting the proliferation of progenitor cells in the embryonic cerebral cortex (*Bedford et al., 2005*; *Yun et al., 2004*). This difference may reflect the different role of the bHLH proteins that are Id4 targets in embryonic versus adult neural lineages. In the embryonic forebrain, NSCs are in a proliferative state and proneural bHLH proteins mostly act to promote neuronal differentiation. Their inactivation by Id4 during development therefore results in a block of differentiation and extended proliferation. In contrast, RGLs in the adult hippocampus are mostly quiescent, Ascl1 is required to promote their activity, and inactivation of Ascl1 by Id4 results in their failure to proliferate.

We also show that Id4 promotes the degradation of the pro-activation factor Ascl1 in RGLs. As Ascl1 protein is only detectable in proliferating RGLs (*Andersen et al., 2014*), we were not expecting the Ascl1 gene to be transcribed by most RGLs including many quiescent cells. We found that despite Ascl1 mRNA being expressed and translated, Ascl1 protein does not accumulate in quiescent RGLs due to its rapid degradation. This surprising finding could be the reason why single cell transcriptomic analysis of hippocampal cells did not identify Ascl1 among the genes differentially expressed between quiescent and active stem cells (*Artegiani et al., 2017*; *Hochgerner et al., 2018*; *Shin et al., 2015*). This non-transcriptional control of a key activation factor is also found, for instance, in satellite stem cells where the bHLH factor MyoD is transcribed in quiescent cells but its translation is inhibited by an RNA-binding protein to prevent stem cell activation (*de Morrée et al., 2017*).

It is well established that Id proteins, including Id4, form non-functional heterodimers with E proteins, which are dimerization partners of tissue-specific bHLH transcription factors such as Ascl1 (*Imayoshi and Kageyama, 2014*; *Ling et al., 2014*; *Patel et al., 2015*; *Sharma et al., 2015*). The three genes encoding for E proteins in mice are all expressed by RGLs, and since they are thought to have redundant functions, it is difficult to investigate their specific contributions to RGL behaviour. Nevertheless, high levels of Ids are expected to result in the sequestration of all E proteins away from functional dimers with Ascl1. Non-dimerized Ascl1 is not able to bind DNA, and this alone could explain why Ascl1 target genes are downregulated in NSCs upon Id4 overexpression or BMP treatment, which increases the expression of all Id proteins. But how does Id4 prevent Ascl1 protein accumulation? Exposure of different cell types to BMPs has been shown to trigger the proteolytic

degradation of Ascl1 (*Shou et al., 1999*; *Viñals et al., 2004*). In lung carcinoma cells, the formation of heterodimers with E47 stabilizes Ascl1, and induction of Id1 by BMP2 sequesters E47, resulting in degradation of the unstable monomeric form of Ascl1 (*Viñals et al., 2004*). We show that Ascl1 is more unstable in quiescent than proliferating hippocampal NSCs and that Ascl1-E47 dimers are disrupted by Id4. Therefore, we propose that a similar mechanism to that in lung carcinoma cells promotes the degradation of Ascl1 when quiescent hippocampal stem cells express high levels of Id4.

To interfere with the function of *Id4* in NSCs and circumvent the compensation by other Id proteins, we have overexpressed E47, which is expected to interact with and titrate all Id proteins. We found that this is indeed sufficient to stabilize Ascl1 protein and promote cell cycle re-entry of BMP-treated NSCs. However, we realize that E47 overexpression might interfere with other factors than Id proteins and Ascl1. Silencing Id4 in quiescent NSCs in culture was confounded by the functional compensation of the other Id proteins, which are able to suppress Ascl1 protein levels when overexpressed. Loss of Id4 in hippocampal RGLs also resulted in increased Id1 and Id3 protein levels, suggesting that Id4 may suppress these proteins in vivo, and that its loss may be partially compensated by the increase in their expression. Since deletion of Id4 has the limitation of functional compensation by other Ids, more refined tools will be required in the future to dissect the specific mechanisms by which Id4 upregulation leads to Ascl1 degradation.

Our transcriptomic analysis suggests that Id4 alone contributes to a large extent to the gene expression program induced by BMP to promote NSC quiescence. Overexpression of Id4 in the absence of BMP induces many of the genes that BMP4 induces, and suppresses many of the genes suppressed by BMP4. Among the genes suppressed by both BMP4 treatment and Id4 overexpression, an important fraction corresponds to cell cycle regulators, including many Ascl1 targets.

Besides regulating the activity of tissue-specific bHLH transcriptional activators such as Ascl1, Id proteins also interact with bHLH transcriptional repressors of the Hes family (*Bai et al., 2007*). Direct interaction of Id2 with Hes1 blocks the autorepressive activity of Hes1 protein resulting in its stable expression at a high level. Therefore, Id proteins promote a switch of the expression pattern of Hes proteins from oscillating, resulting in oscillatory expression of target genes such as Ascl1, to stably high, resulting in constant repression of these targets (*Bai et al., 2007*; *Boareto et al., 2017*; *Sueda et al., 2019*). However, we find that *Ascl1* is transcribed in most quiescent RGLs, indicating that Hes proteins do not repress *Ascl1* transcription in these cells. Id4 has been shown to inhibit the action of Id1-3 proteins by interacting with stronger affinity with them than with other binding partners (*Patel et al., 2015*; *Sharma et al., 2015*). This non-canonical role of Id4 has been proposed to explain that Id4 promotes the proliferation of embryonic neural progenitors by blocking the interaction of Id1-3 with Hes proteins and thus promoting oscillations of Hes proteins – and consequently of Ascl1 – and progenitor proliferation (*Bedford et al., 2005*; *Boareto et al., 2017*; *Yun et al., 2004*). Id4 might therefore maintain Ascl1 transcription – separately from its role in eliminating Ascl1 protein – by interfering with the role of other Id proteins in stabilising Hes protein expression. In support of this model, *Id4* deletion in RGLs in vivo results not only in upregulation of other Id proteins but also in a reduction of *Ascl1* transcript levels. The function that we have identified for Id4 in the hippocampus is also distinct from the role reported for other Id factors in the adult V-SVZ, where Id1 and Id3 promote stem cell self-renewal (*Nam and Benezra, 2009*) and Id1-3 maintain stem cell function by keeping stem cells adherent to their niche environment (*Niola et al., 2012*).

Id4 is not the only Id protein expressed in hippocampal RGLs, as Id1 is also expressed in nearly half of them. Nevertheless, Id4's role in the regulation of stem cell quiescence is clearly different from that of Id1. While Id4 expression is restricted to RGLs that are quiescent and express low levels or no Ascl1 protein, Id1 protein is found in proliferating RGLs, many of which also express Ascl1 protein, suggesting that contrary to Id4, Id1 at the level it is expressed in RGLs in homeostasis does not promote stem cell quiescence or the degradation of Ascl1, although the same factor may have the potential to promote Ascl1 degradation when expressed at higher levels, that is when overexpressed in cultured NSCs. In agreement with this, Id1 has recently been shown to have a role in the activation of hematopoietic stem cells upon stress signals (*Singh et al., 2018*). We have addressed the effect of loss of *Id1* from RGLs by examining *Smad4cKO* mice, where Id1 expression in RGLs is greatly reduced while Id4 expression is unaffected (*Blomfield et al., 2018*). Loss of *Smad4* did not affect RGLs, indicating that Id1 is not required to suppress Ascl1 expression or RGL proliferation. It is unclear why Id1, which has been shown to dimerize with E proteins and promote Ascl1 degradation in another cell type (*Viñals et al., 2004*) has no such effect in hippocampal stem cells, but this

may be the result of its relative expression levels in RGLs. Id3 expression, on the other hand, is mostly restricted to quiescent RGLs and is therefore a better candidate to compensate for the loss of Id4 in quiescent cells.

While Id4 is expressed in the vast majority of quiescent RGLs, *Id4* deletion results in loss of quiescence of only a fraction of them at both 5d and 30d post-deletion, suggesting that compensatory mechanisms operate to blunt the *Id4$^{cKO}$* phenotype in RGLs. Thus, other Ids - in particular Id3, because of its expression pattern in quiescent RGLs - or unrelated factors might compensate partially for the loss of *Id4*. An increase in the repression of *Ascl1* transcription, due to increased Hes-Id protein interactions and Hes protein stabilization, might also contribute to the blunting of the *Id4$^{cKO}$* phenotype. In support of this, *Ascl1* transcription was lower in *Id4$^{cKO}$* mice as early as 5 days after *Id4* deletion. We have previously shown that the ubiquitin ligase Huwe1 degrades Ascl1 in proliferating RGLs and allows a fraction of these cells to return to quiescence (*Urbán et al., 2016*). Huwe1 is expressed in quiescent RGLs (*Urbán et al., 2016*) and although Id4 might mask its role in degrading Ascl1 in these cells, it might be able, in the absence of Id4, to eliminate excess Ascl1 and maintain RGL quiescence.

Given the important role of Id4 in maintaining RGL quiescence, it seems likely that its expression is regulated by niche signals to control RGL activity. *Id* genes, including *Id4*, are well known effectors of BMP signalling in neural cells and other cell types (*Ling et al., 2014*; *Patel et al., 2015*; *Samanta and Kessler, 2004*). *Smad4* deletion strongly reduces Id1 but not Id4 levels in RGLs, indicating that Id4 diverges from other Id proteins not only in its activity but also in the regulation of its expression (*Blomfield et al., 2018*). *Id4* has been shown to be directly regulated by Notch signalling in embryonic neural progenitors (*Li et al., 2012*) and in adult hippocampal stem cells (*Zhang et al., 2019*) but we found that Id4 expression persists in RGLs lacking the essential Notch signalling effector *RBPJk* (*Blomfield et al., 2018*). Id4 expression is only mildly affected in RGLs lacking both *Smad4* and *RBPJk*, indicating that additional pathways beside BMP-Smad4 and Notch-Rbpjk promote RGL quiescence by maintaining Id4 expression.

Id4 is expressed in most quiescent RGLs but it is sharply downregulated in active RGLs. Indeed, it is one of the most differentially expressed genes in quiescent versus active stem cells, both in vivo and in NSC cultures (*Shin et al., 2015* and this paper). Down-regulation of Id4 is crucial for RGLs to produce sufficient levels of Ascl1 protein to leave the quiescent state and become active, emphasizing the importance of this gene in the maintenance of RGL quiescence. An important aim of future research will be to identify the niche signals that induce Id4 expression in quiescent RGLs and reduce its expression in active cells.

## Materials and methods

Contact for reagent and resource sharing: François Guillemot (Francois.guillemot@crick.ac.uk).

### Experimental model and subject details

#### Mouse models

All procedures involving animals and their care were performed in accordance with the guidelines of the Francis Crick Institute, national guidelines and laws. This study was approved by the Animal Ethics Committee and by the UK Home Office (PPL PB04755CC). Mice were housed in standard cages under a 12 hr light/dark cycle, with ad libitum access to food and water. All experimental mice were of a mixed genetic background. Founder mice were bred to MF1 mice, and then backcrossed to littermates of the F1 generation. Experimental strains used:

- Ascl1Venus (Ascl1$^{tg1(venus)Rik}$) mice, originally reported by *Imayoshi et al. (2013)*.
- Ascl1KiGFP (Ascl1$^{tm1Reed}$) mice, originally reported by *Leung et al. (2007)*.
- Id4flx mice, originally reported by *Best et al. (2014)*.

In order to generate mice with a hippocampal stem cell-specific, tamoxifen-inducible recombination, plus a tdTomato reporter of recombination, GLAST-CreERT2 (Slc1a3$^{tm1(cre/ERT2)Mgoe}$) (*Mori et al., 2006*) mice were crossed with Gt(ROSA)26Sor$^{tm9(CAG-tdTomato)Hze}$ (tdTomato) mice, originally reported by *Madisen et al. (2010)*. These mice were further crossed to the Id4flx strain to generate inducible conditional Id4 knockout mice with a tdTomato reporter of recombination. Id4flx mice crossed with Glast-CreERT2 and Rosa26-floxed-stop-YFP (RYFP; Gt(ROSA)26Sor$^{tm1(EYFP)Cos}$)

mice (*Srinivas et al., 2001* were also kindly provided by B. Rocamonde and E. Huillard. Analyses of these mice are reported in *Blomfield et al. (2018)* and were also used in the quantification of data shown in *Figure 6D and J* and S6H.

Both male and female mice were used for all in vivo genetic studies. Experimental groups were a mix of animals from different litters for each particular strain. All mice were injected with tamoxifen at postnatal day 60 ± 2, and brain tissue collected by transcardial perfusion at 2, 5, 10 or 30 days after the first injection.

## Primary Cell Cultures

For the derivation of adult hippocampal stem cell lines, 7–8 week old mice were sacrificed and the dentate gyrus dissected (previously described by *Walker et al., 2013*). Cultures were amplified as neurospheres for two passages before dissociation to adherent cultures. Cells were propagated in basal media (DMEM/F-12 + Glutamax (Invitrogen 31331–093) + 1x Neurocult Supplement (Stem Cell Technologies, 05701) + 1x Penicillin-Streptomycin (ThermoFischer Scientific, 15140)+ 2 µg/ml Laminin (Sigma, L2020) + 20 ng/ml FGF (Peprotech, 450–33) + 5 µg/mL Heparin (Sigma, H3393-50KU). Cells were incubated at 37°C, 5% $CO_2$.

The wildtype adult hippocampal neural stem cell line (AHNSC line #5) was derived from a single male WT/RYFP mouse. AHNSC Ascl1Venus cell line was derived from a single male Ascl1$^{wt/Venus}$ mouse. Huwe1 is X-linked, therefore AHNSC Huwe1flx cell line was derived from a male Glast-CreERT2$^{wt/wt}$; Huwe$^{fl/Y}$;Rosa$^{YFP/YFP}$ mouse.

## Method details

### Tamoxifen administration

To induce activation of CreERT2 recombinase, 2 mg (57–67 mg/Kg) of tamoxifen (Sigma, T5648) was administered intraperitoneally (ip) to mice at postnatal day 60 (P60), at the same time each day for five consecutive days. For in situ hybridization experiments, Glast-CreERT2;tdTomato (Ai19) mice received a single injection at postnatal day 60 + /- 2, and brain tissue collected by transcardial perfusion 48 hr later.

### Tissue preparation and immunofluorescence

Mice were transcardially perfused with phosphate-buffered saline (PBS) for 3mins, followed by 4% paraformaldehyde (PFA) in PBS for 12mins. Brains were post-fixed for 2 hr in 4% PFA at 4°C and washed with PBS. Brains were coronally sectioned at a thickness of 40 µm using a vibratome (Leica).

For in situ samples, mice were perfused with PBS for 3mins, followed by 10% neutral buffered formalin (NBF) for 12mins. Brains were post-fixed in 10% NBF at room temperature for 16–32 hr, and then washed with 70% EtOH. Brains were paraffin embedded, and coronally sectioned at a thickness of 5 µm.

Cultured cells were fixed with 4%PFA in PBS for 10mins at room temperature, and washed with PBS.

For immunofluorescence of tissue, samples were blocked with 10% normal donkey serum (NDS) in 1%Triton-PBS for 2 hr at room temperature with rocking. Fixed cells were blocked with 10%NDS in 0.1%Triton-PBS for 1 hr at room temperature. Primary antibodies were diluted in 10%NDS in 0.1% Triton-PBS, and incubated with samples overnight at 4°C with rocking. Following 3 × 0.1% Triton-PBS washes, samples were incubated with secondary antibodies diluted in 10%NDS in 0.1%Triton-PBS for 2 hr at room temperature with rocking. Following 3 × 0.1% Triton-PBS washes, samples were incubated with 1 µg/mL DAPI (Sigma, D9542) in 1:1 PBS:$H_2O$ for 30mins at room temperature with rocking. Primary and secondary antibodies and dilutions are listed in in *Supplementary file 1*.

EdU was detected following secondary antibody incubation, using Click-iT EdU Alexa Fluor 647 Imaging Kit (Invitrogen, C10340), following manufacturer's instructions.

### RNA in situ hybridization

For RNA in situ hybridization, the RNAscope Multiplex Fluorescent Reagent Kit V2 (ACD Bio-Techne, 323110) was used with NBF fixed-paraffin embedded 5 µm sections, and stained according to the standard company protocol. Target retrieval was performed for 15mins, and Protease Plus treatment was carried out for 30mins. For dual RNAscope-immunofluorescence, following the development of

HRP-C3 signal and wash steps, slides were washed in distilled $H_2O$, and washed $3 \times 5$ mins in 0.1% Triton-PBS at room temperature. Slides were then processed for immunofluorescence as described above. Probes, fluorophores and dilutions listed in *Supplementary file 1*.

## Microscopic analysis

All images were acquired using an SP5 confocal microscope using the 40X objective lens (Leica). For cell culture immunofluorescence, three random regions of each coverslip were imaged with a z-step of 1 µm. For adult tissue immunofluorescence, both left and right dentate gyri of every twelfth 40 µm section along the rostrocaudal length of the DG were imaged, with a z-step of 1 µm through the whole 40 µm section. For quantification of %+ RGLs, at least 200 RGLs in each of at least three mice for each genotype were quantified.

RGLs were identified based on their characteristic morphology (nucleus in the subgranular zone, radial process projecting through the molecular layer) and positive labeling with GFAP and GFP in the case of Glast-CreERT2;RYFP recombined cells, or tdTomato positivity in the case of Glast-CreERT2;tdTomato recombined cells.

## Cell treatments, constructs and transfection

For culturing adult hippocampal NSCs in proliferation conditions, cells were grown in basal media (DMEM/F-12 + Glutamax (Invitrogen, 31331–093)) + 1x N2 supplement (R and D Systems, AR009) + 1x Penicillin-Streptomycin (ThermoFischer Scientific, 15140) + 2 µg/ml Laminin (Sigma, L2020) + 5 µg/mL Heparin + 20 ng/ml FGF2 (Peprotech, 450–33). To induce quiescence, cells were plated into flasks or onto coverslips in proliferation conditions and allowed to adhere overnight. Media was replaced the next day with basal media or basal media plus 20 ng/mL recombinant mouse BMP4 (R and D Systems, 5020 BP), and cultured for 72 hr at 37°C, 5% $CO_2$.

To test that BMP4-induced cells could reactivate and differentiate, BMP4-treated cells were detached from their flask using Accutase (Sigma, A6964) and re-plated into proliferation conditions, and fixed at 24 hr, 48 hr and 72 hr post-reactivation. To differentiate the cells, following 72 hr reactivation, cells were cultured in the presence of 10 ng/mL FGF2% and 2% foetal bovine serum, for 72 hr.

In order to visualize S-phase, EdU (Invitrogen, C10340) was added to the media of cells in culture 1 hr prior to fixation.

To delete Huwe1 in NSCs derived from Huwe1 floxed transgenic mice, *Huwe1*$^{fl/fl}$ NSCs were induced to quiescence with 20 ng/mL FGF2 plus 20 ng/mL BMP4 for 72 hr, then transduced with either control adenovirus (Adeno-empty) or adenovirus expressing Cre recombinase (Adeno-Cre) at a concentration of 100MOI and cultured for a further 6 days to ensure complete degradation of the very stable Huwe1 protein.

To inhibit the proteasome and to measure the half-life of Ascl1, cells were grown on 10 cm diameter dishes for 72 hr in supplemented basal media with either just 20 ng/mL FGF2 or FGF2 + 20 ng/mL BMP4. For proteasomal inhibition, cells were treated with either 10 µM MG132 (Sigma, M7449) or an equal volume of DMSO (Sigma), for 0, 5, 10, 30, 60 or 120mins. To measure Ascl1 half-life, cells were treated with either 100 µg/mL cycloheximide (Sigma, C4859) or an equal volume of DMSO for 0, 15, 30, 60, 120 and 240mins.

For overexpression of Id1, Id2, Id3, Id4 and E47 in NSCs, $5 \times 10^6$ cells per construct were nucleofected with 10 µg DNA using the Amaxa mouse neural stem cell nucleofector kit (Lonza, VPG-1004) and Amaxa Nucleofector II (Lonza), using the program A-033, according to manufacturer's instructions. The pcβ-Id4-FLAG construct, pBABE-empty and pBABE-N-FLAG-ID-puro constructs were kind gifts from M. Israel (*Rahme and Israel, 2015*). The E47 expression construct was generated by cloning E47 into pCAGGS-IRES-GFP via EcroRV/Xho1. In order to FACS sort Id4-transfected cells, cells were co-transfected with an empty pCAGGS-IRES-GFP construct at half the concentration of pcβ-Id4-FLAG, to increase the likelihood of GFP+ cells also being Id4+. For FACS and subsequent RNA-seq analysis, FGF2 and FGF2+BMP4 control samples were nucleofected with pmaxGFP vector from the Amaxa kit (Lonza, VPG-1004). Following transfection, cells were plated into flasks and onto glass coverslips, in supplemented basal media, and incubated for 48 hr at 37°C, 5% $CO_2$. Cells transfected with Id1, Id2, Id3 and Id4 were cultured in the presence of 20 ng/mL FGF2; cells transfected with E47 were cultured in the presence of both 20 ng/mL FGF2 and 20 ng/mL BMP4.

For overexpression of Ascl1, E47 and Id4 in 293 T cells, $2 \times 10^6$ cells per construct were transfected using Lipofectamine LTX (ThermoFisher Scientific, 15338100) according to manufacturer's instructions. 10 µg of each construct was transfected separately and cells were cultured for 72 hr following transfection. The pBABE-empty and pBABE-N-FLAG-Id4 constructs were a kind gift from M. Israel (*Rahme and Israel, 2015*). The pCAGGS-E47-IRES-GFP vector and pCAG-Ascl1-V5 vector (produced by D.v.d.B) were used to overexpress E47 and Ascl1-V5, respectively.

To silence the expression of Id1, Id2, Id3 and Id4 in quiescent NSCs, NSCs were treated for 72 hr with 20 ng/mL FGF2 and 20 ng/mL BMP4 and then nucleofected using the Amaxa mouse neural stem cell nucleofector kit (Lonza, VPG-1004) and Amaxa Nucleofector II (Lonza), using the program A-033, according to manufacturer's instructions. NSCs were transfected with a cocktail of three siRNAs at 20 nM each, or 60 nM control scrambled siRNA, or 60 nM HPRT-targeting siRNA as a positive control (Origene). To silence Id4 in combination with Id1, Id2 or Id3, 30 nM of Id4 'A' siRNA was co-transfected with 30 nM of siRNA 'A' targeting Id1, 2 or 3. Following transfection, cells were plated into P6 wells and onto glass coverslips, in supplemented basal media plus 20 ng/mL FGF2 and 20 ng/mL BMP4, and incubated for 48 hr at 37°C, 5% $CO_2$.

## FAC sorting

FACS tubes were pre-coated with 5%BSA-PBS at 37°C for at least 30mins prior to sorting. Cells were dissociated from flasks using Accutase (Sigma) and centrifuged at 0.3RCF for 5mins. Cell pellets were resuspended in 750 µL recovery media (5%BSA-PBS + 20 ng/ml FGF + 1 µg/mL Heparin). 1 µL propidium iodide was added to cell suspensions to check for cell viability. Cells were sorted on a FACS Aria III machine, into recovery media. Both GFP positive and negative cells were recovered into separate tubes.

## RNA extraction, cDNA synthesis and QPCR

For FACS experiments, cells were lysed using Qiagen lysis buffer. For all other experiments, cells were lysed with Trizol reagent. RNA was extracted using RNeasy Mini Kit (Qiagen, 74104) or Direct-zol RNA MiniPrep Kit (Zymo Research, R2052), according to manufacturer's instructions. cDNA was synthesized using the High Capacity cDNA Reverse Transcription Kit (Applied Biosystems, 4387406) following manufacturer's instructions. Gene expression level was measured using TaqMan Gene expression assays (Applied Biosystems) and quantitative real-time PCR carried out on a QuantStudio Real-Time PCR system (ThermoFisher). Gene expression was calculated relative to endogenous controls Gapdh and ActinB, and normalized to the expression of one control sample in each group, to give a ddCt value. All samples were measured in technical duplicates for each QPCR run and averaged. At least three biological replicates were performed for each condition.

## RNA sequencing and analysis

RNA concentration was quantified using the Qubit dsDNA BR/HS Assay Kit. A KAPA mRNA Hyper-Prep Kit (for Illumina) (KAPA Biosystems, Wilmington, MA, USA) was used with 1000 ng of RNA diluted to a final volume of 50 µl. Each RNA sample was captured with 50 µl of capture beads at 65°C for 2 min and 20°C for 5 min. For the second capture, 50 µl of RNase free water was used at 70°C for 2 min and 20°C for 5 min. Captured RNA was subjected to the KAPA Hyper Prep assay: end-repair, A-tailing, and ligation by adding 11 µl of Fragment, Prime and Elite Buffer (2X). To obtain a distribution of 200–300 bp fragment on the library, the reaction was run for 6 min at 94°C. cDNA synthesis was run in two steps following manufacturer's instructions. The ligation step consisted of a final volume of 110 µL of the adaptor ligation reaction mixture with 60 µL of input cDNA, 5 µL of diluted adaptor and 45 µL of ligation mix (50 µL of ligation buffer+ 10 µL of DNA ligase). The Kapa Dual-Indexed Adapters (KAPA Biosystems-KK8720) stock was diluted to 7 µM (1.5 mM or 7 nM) to get the best adaptor concentration for library construction. The ligation cycle was run according to manufacturer's instructions. To remove short fragments such as adapter dimers, 2 AMPure XP bead clean-ups were done (0.63 SPRI and 0.7SPRI). To amplify the library, 7 PCR cycles were applied to cDNA KAPA HP mix. Amplified libraries were purified using AMPure XP. The quality and fragment size distributions of the purified libraries was assessed by a 2200 TapeStation Instrument (Agilent Technologies, Santa Clara, CA, USA).

Libraries were sequenced with Hiseq4000 (Illumina), 50 bp paired-end reads for sequencing proliferating vs quiescent NSCs; 75 bp single-end reads for Id4/E47 overexpressing NSCs, with a depth of $30 \times 10^6$ reads.

The quality of RNA sequence reads was evaluated using FastQC (version 0.11.2) (*Andrews, 2010*). Low quality reads and contaminants (e.g. sequence adapters) were removed using Trimmomatic (version 0.32) (*Bolger et al., 2014*). Sequences that passed the quality assessment were aligned to the mm10 genome using tophat2 (version 2.0.14) (*Kim et al., 2013*), with bowtie2 (version 2.1.0) (*Langmead and Salzberg, 2012*) or for the quiescent NSC RNAseq data set, Cufflinks (*Trapnell et al., 2010*). Transcript abundance level (transcript count) was generated using HTSeq (version 0.5.3p9) (*Anders et al., 2015*). The transcript counts were further processed using R software environment for statistical computing and graphics (version 3.4.0). Data normalisation, removal of batch effect and other variant was performed using EDASeq R package (*Risso et al., 2011*) and RUVseq package (Remove Unwanted Variation from RNA-Seq package) (*Risso et al., 2014*). Differential expression was performed using edgeR R package (*Robinson et al., 2010*), using the negative binomial GLM approach, or for the quiescent NSC RNAseq data set, Cuffdiff (version 7) (*Trapnell et al., 2013*). Differentially expressed genes with false discovery rate (FDR <= 0.05, Benjamini-Hochberg multiple testing correction), expression level in control samples > 1 CPM (counts per million) or > 1 FPKM (fragments per kilobase of transcript per million mapped reads) for the quiescent NSC RNAseq data set, and log fold change > 1 were retained and used for further processing, gene ontology and pathway analysis.

## Protein purification, western blot and co-immunoprecipitation

WT and Ascl1-Venus NSCs were cultured in 10 cm diameter dishes, in either proliferation or quiescent conditions for 72 hr. Media was refreshed after 40 hr to ensure constant BMP signalling. Cells were then washed with ice-cold PBS, and scraped in Lysis Buffer (ThermoFischer Scientific, 87788) + 1x Protease inhibitor cocktail (ThermoFischer Scientific, 87786) + 1 x EDTA (ThermoFischer Scientific, 87788) + 1x Phosphatase inhibitor cocktail (ThermoFischer Scientific, 78420). Cells were lysed at 4°C for 20 min under rotation and then centrifuged at 13,000 RPM at 4°C for 20mins and the pellet discarded. The supernatant was analyzed either by western blot or subject to immunoprecipitation or in vitro competition-binding assay. For western blot analysis, the supernatant was mixed with 1x Laemmli sample buffer (Sigma, S3401-10VL) and incubated at 95°C for five mins. For immunoprecipitation experiments, antibodies were added to cell lysate supernatants and incubated at 4°C for 2 hr under rotation. As controls, mouse anti-V5-tag or rabbit anti-HA-tag antibodies were used under the same conditions. Sepharose coupled to protein G (Sigma, P3296) was blocked with 5% BSA-PBS for 2 hr at 4°C under rotation. After several washes with PBS, it was then added to the lysate-antibody suspension and incubated for 2 hr at 4°C under rotation. After this period, Sepharose beads were washed with lysis buffer five times, then suspended in an equal volume of Laemmli sample buffer and incubated at 95°C for five mins. For the in vitro competition-binding assay, E47 and Id4 proteins were detected by combining the cell lysates in the ratio of 1(Ascl1-V5):1(E47):1(pBABE-Id4 or pBABE-empty). For the detection of Ascl1(V5), the cell lysates were combined in the ratio of 1(Ascl1-V5):1(E47):0.5(pBABE-Id4 or pBABE-empty). Both combinations were then submitted to immunoprecipitation with anti-E47 and anti-V5 antibodies, or rabbit anti-HA-tag antibody as a control. Samples were run in polyacrylamide gel at 120V, after which they were transferred onto a nitrocellulose membrane. Filters were then saturated with 5% BSA in TBS-Tween or 5% milk TBS-Tween and incubated with the antibodies. Detection was performed using ECL western blotting reagents (Sigma, GERPN2106).

## Quantification and statistical analysis

To measure immunofluorescence intensity, the nucleus of each identified RGL was manually outlined based on DAPI staining, and the average pixel value of the channel of interest was measured using FIJI software. Every value was normalized to the background level measured in a negative nucleus in the same z-plane as each RGL. At least 200 RGLs in each of at least three mice were quantified for each protein. For in vitro ICC quantification, average pixel intensity for each channel was measured for the area of each nuclei, using FIJI software. For each experiment, at least 100 cells were quantified across at least three biological replicates. To generate the arbitrary units (A.U.) for both in vivo

and in vitro IHC, all the values within a sample were made relative to the average of the control, and expressed as a %, with the average of control being 100%. For quantification of RNAscope staining, the number of 'dots' in each identified RGL nucleus were counted for each probe. In addition, the average pixel intensity in and around each RGL nucleus was measured for each probe, using FIJI. 100 RGLs were quantified across five mice. For analysis of Id4 and E47 nucleofected cells, Id4+ or GFP+(E47) cells were identified by immunostaining for Id4 or GFP respectively, and positive cells compared to negative, non-transfected cells within the same coverslip. Cell counts were done from at least 3 coverslips from three biological replicates. For quantifying RGL density, the DG volume was calculated by multiplying the length, height and depth of the SGZ imaged in mm, and the number of stem cells counted expressed per mm$^3$. All data were analyzed with masking of genotype/group to avoid bias.

For quantification of WB and IP assays, films were scanned and, if appropriate, subjected to band densitometry and quantification using Image J software (RRID:SCR_002285). Each band value was normalized according to the background of the filter and its loading control.

The appropriate sample size ('n') was determined based on previous experiments of identical characteristics from our previous publications (Andersen et al.; Urban et al.) and similar published data from other groups, using a minimum of 3 mice per condition for in vivo experiments, and a minimum of triplicate for in vitro experiments. Throughout this paper, 'technical replicate' refers to the same sample being tested multiple times; 'biological replicate' refers to independent biological samples. All data collected were included, as variation was considered within expected ranges and variances were non-significant.

Statistical analyses were conducted using the GraphPad Prism seven software (RRID:SCR_002798) using a two-sample unpaired t test assuming Gaussian distribution for the comparison of two conditions; paired t test was used for *Figure 4E–I,L–N*, where the control and treatment conditions for each biological replicate were performed on cultures in parallel; or ordinary one-way ANOVA followed by Tukey's multiple comparisons test, for the pairwise comparison of three conditions. All error bars represent the mean ± SEM. Significance is stated as follows: $p > 0.05$ (ns), $p < 0.05$ (*), $p < 0.01$ (**), $p < 0.001$ (***), $p < 0.0001$ (****), confidence intervals of 95%. Statistical details of each experiment can be found in the figure legend. n represents number of independent biological repeats.

## Acknowledgements

We gratefully acknowledge Lan Chen for technical support, Rekha Subramaniams and Nicholas Chisholm for managing the mouse colony, the Advanced Sequencing Facility, Bioinformatics and Biostatistics Facility, Experimental Histopathology Facility and Flow Cytometry Facility of the Francis Crick Institute for their help and support, Jane Johnson, Ryoichiro Kageyama, Ana Lasorella, Randall Reed, Jane Visvader for providing mice, Gilbert Rahme, Matthew Havrda and Mark Israel for plasmids, Hiyaa Ghosh for advice on Tcf4 immunostaining, Runrui Zhang and Verdon Taylor for sharing data and manuscript before publication and members of the Guillemot lab for discussions. NU was supported by a fellowship from the Francis Crick Institute and is currently supported by IMBA (Austrian Academy of Science); BR was supported by H2020-MSCA-IF-2014; DvdB was supported by a Marie Curie Fellowship, project 799214; EH was supported by Ligue Nationale contre le Cancer, Fondation ARC pour la recherche sur le cancer (PJA 20131200481, PJA 20151203259) and FP7 Marie Curie CIG. This work was supported by the Francis Crick Institute, which receives its funding from Cancer Research UK (FC0010089), the UK Medical Research Council (FC0010089) and the Wellcome Trust (FC0010089), by the UK Medical Research Council (project grant U117570528 to FG) and by the Wellcome Trust (Investigator Award 106187/Z/14/Z to FG). The authors declare no conflict of interest. Supplement contains additional data.

## Additional information

### Competing interests

François Guillemot: Reviewing editor, *eLife*. The other authors declare that no competing interests exist.

## Funding

| Funder | Grant reference number | Author |
| --- | --- | --- |
| Francis Crick Institute | FC0010089 | Francois Guillemot |
| Medical Research Council | U117570528 | Francois Guillemot |
| Wellcome | 106187/Z/14/Z | Francois Guillemot |
| H2020 Marie Skłodowska-Curie Actions | Project 799214 | Debbie LC van den Berg |
| Ligue Contre le Cancer | PJA 20131200481 | Emmanuelle Huillard |
| Ligue Contre le Cancer | PJA 20151203259 | Emmanuelle Huillard |
| H2020 Marie Skłodowska-Curie Actions | H2020-MSCA-IF-2014 | Brenda Rocamonde |

The funders had no role in study design, data collection and interpretation, or the decision to submit the work for publication.

## Author contributions

Isabelle Maria Blomfield, Conceptualization, Data curation, Formal analysis, Validation, Investigation, Visualization, Methodology, Writing—original draft, Project administration, Writing—review and editing; Brenda Rocamonde, Emmanuelle Huillard, Resources, Collaborative discussion of data; Maria del Mar Masdeu, Data curation, Formal analysis, Validation, Investigation, Visualization; Eskeatnaf Mulugeta, Software, Formal analysis, Visualization; Stefania Vaga, Software, Formal analysis; Debbie LC van den Berg, Software, Formal analysis, Visualization, Methodology; François Guillemot, Conceptualization, Resources, Supervision, Funding acquisition, Methodology, Writing—original draft, Project administration, Writing—review and editing; Noelia Urbán, Conceptualization, Data curation, Formal analysis, Supervision, Validation, Investigation, Visualization, Methodology, Writing—original draft, Project administration, Writing—review and editing

## Author ORCIDs

Isabelle Maria Blomfield (iD) http://orcid.org/0000-0003-4412-0226
Debbie LC van den Berg (iD) http://orcid.org/0000-0001-6026-8808
François Guillemot (iD) https://orcid.org/0000-0003-0432-5067
Noelia Urbán (iD) https://orcid.org/0000-0003-4813-9274

## Ethics

Animal experimentation: All procedures involving animals and their care were performed in accordance with the guidelines of the Francis Crick Institute, national guidelines and laws. This study was approved by the Animal Ethics Committee and by the UK Home Office (PPL PB04755CC). All surgery was performed under terminal pentobarbital anaesthesia, and every effort was made to minimise suffering.

## Decision letter and Author response

Decision letter https://doi.org/10.7554/eLife.48561.027
Author response https://doi.org/10.7554/eLife.48561.028

## Additional files

### Supplementary files

• Source data 1. Transparent reporting summary statistics.
DOI:
• Supplementary file 1. Key resources table.
DOI: https://doi.org/10.7554/eLife.48561.021
• Transparent reporting form DOI: https://doi.org/10.7554/eLife.48561.024

## Data availability

Sequencing data have been deposited in GEO under accession code GSE116997.

The following dataset was generated:

| Author(s) | Year | Dataset title | Dataset URL | Database and Identifier |
|---|---|---|---|---|
| Blomfield IM, Mulugeta E, Vaga S, van den Berg DLC, Guillemot F, Urban N | 2018 | Id4 maintains quiescence of adult hippocampal stem cells | https://www.ncbi.nlm.nih.gov/geo/query/acc.cgi?acc=GSE116997 | NCBI Gene Expression Omnibus, GSE116997 |

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
