## [Decision Letter]

Thank you for submitting your article "Id4 promotes the elimination of the pro-activation factor Ascl1 to maintain quiescence of adult hippocampal stem cells" for consideration by *eLife*. Your article has been reviewed by two peer reviewers, and the evaluation has been overseen by Gary Westbrook as the Senior and Reviewing Editor. The reviewers have opted to remain anonymous. The reviewers have discussed the reviews with one another and the Reviewing Editor has drafted this decision to help you prepare a revised submission.

Summary:

This manuscript adds important details to understanding the ways that dentate stem cells' quiescence is regulated. The reviewers were positive about the manuscript, but raised several issues that must be addressed in a revision of which the first point is the most critical. The original reviewers are included below.

1) The role of other Id's. Is there redundancy or are there specific molecular functions of Id4. It is essential to include or exclude a role for Id1-3, given that the focus of the manuscript (and the title) is on Id4. (point 2, 3, 4 of reviewer 1, point 1 of reviewer 2).

2) The role of Hes family genes (point 1 of reviewer 1).

3) The cell cycle experiments would strengthen their conclusions but are non-essential (point 2 of reviewer 2).

Reviewer #1:

In the manuscript "Id4 promotes the elimination of the pro-activation factor Ascl1 to maintain quiescence of adult hippocampal stem cells", Blomfield et al. described the role of Id4 on regulating quiescence of neural stem cells (NSCs) in adult dentate gyrus. The importance of Ascl1 in activating quiescent NSCs has been demonstrated, and the authors presented data show that Id4 functions to reduce Ascl1 protein levels in the NSCs, thereby contributing to maintaining the quiescent state of NSCs. In addition, they show that Id4 expression is upregulated by BMP stimulation, a signaling pathway that promotes NSCs quiescence.

The authors suggest that BMP stimulation decreases Ascl1 protein thorough Id4, which competitively binds to the binding partner of Ascl1 leading to the elimination of Ascl1 protein and maintaining the quiescence of NSCs. Although most of their functional analysis were done in vitro, their analysis is well established, and the authors provide interesting insights into Id4 function in the post-translating regulation of Ascl1 and regulating quiescent NSCs of DGs. However, there are several concerns in the functional analysis of Id4 and Ascl1.

1) The authors indicated that Ids are upregulated upon BMP stimuli, and that Id4 negatively regulates the function of Ascl1 by destabilizing it. However, it is well-known that BMP also increases the expression of Hes family genes in some contexts, which could negatively regulate the function of Ascl1 and its expression. From their data that BMP stimulation does not decrease the mRNA levelS of Ascl1, it seems likely that Hes genes are not upregulated by BMP in their system, or that the expression of Hes does not influence the expression of Ascl1 in vitro. However, given all these caveats, it would be helpful if the authors examined the expression of Hes family genes in their qPCR or RNA-seq data.

2) In Figure 6, they showed that Id1 and Id3 expression are upregulated in RGLs of *Id4^cKO^* mice, supporting their idea that Id1 and Id3 compensate in part for Id4 function. However, the authors did not provide any data that Id1 and Id3 work similarly as Id4 for maintaining quiescence of NSCs and for the destabilization of Ascl1 (using their in vitro system). It is interesting to know if each Ids has different specific function or works similarly/together in NSCs. In this case, because BMP also upregulates the expression of Id1/3 not only Id4, I wonder if maintaining quiescent state and destabilization of Ascl1 is unique function of Id4, or if Id1/3 also can play similar role as Id4 in their culture system.

3) The authors show that BMP signaling decreases the Ascl1 protein levels, and they suggested that it is mediated by Id4. To support this, the author showed that Id4 overexpression reduced Ascl1 protein amount by immunofluorescence staining. However, it is still possible that BMP signaling decreases Ascl1 protein using other mechanisms in addition to Id4. Given that, I wonder if they could perform some experiments to demonstrate how BMP signaling affects Ascl1 stability and NSC proliferation when Id4 protein is absent or reduced in their in vitro system. There are ways perhaps to simultaneously regulate Id1, 3 and 4 in this system.

4) They showed that Id4 competes with Ascl1 for binding with E47 in vitro. From this aspect, they assumed that the amount of Id4 increases monomeric of Ascl1 thereby destabilizing Ascl1. However, this mechanism isn't really clear. Furthermore, the reports they cited in the Discussion section showed that Id4 function is different from the other Ids. The binding affinity of Id4 to E47 is less than the one in other Ids (Sharma et al., 2015). This makes me wonder that destabilization of Ascl1 by Id4 is due to other mechanisms. The fact that deletion of Id4 increases the amount of Id1/3 without changing their protein level suggests that Id4 may also negatively regulate the protein stability of Id1/3. Because the paper also reported that Id4 makes heterodimer with other Ids to inhibit their function, I wonder that if they could also address the possibility that Id4 makes heterodimer with Ascl1 negatively regulates its function and leading to destabilization.

Reviewer #2:

The molecular mechanisms that control the transition between quiescent and proliferation states remain poorly understood. This is particularly important in the adult dentate gyrus of the mammalian hippocampus where adult neural stem cells (radial glial cells or RGLs) remain largely quiescent which is essential for their long-term self-renewal. Previously the Guillemot lab showed that the bHLH transcription factor ASCL1 promotes activation of RGL cells and generation of new neurons (Andersen et al., 2014). The Guillemot lab also showed that stabilization of ASCL1 protein by inactivation of the E3 ligase Huwei1 results in over-proliferation of adult neural stem cells and prevents their return to quiescence (Urban et al., 2016). Building on this early work, the authors now show: (1) ASCL1 is transcriptionally active in both quiescent and proliferating RGLs but only activated RGLs express ASCL1 protein, (2) Id4 is a candidate factor to reduce ASCL1 protein in quiescent RGLs, (3) Id4 sequesters E47 and ASCL1 becomes unstable, (4) Id4 is required to reduce ASCL1 levels in quiescent RGLs in adult mouse hippocampus.

These data are rigorously performed and the authors use both loss- and gain-of-function and in vitro and in vivo approaches. The single cell RNA in situ hybridization, biochemical assays using tagged constructs, and conditional Id4 knockout mice are particularly compelling. Although Id4's regulation of ASCL1 is not new, the function of Id4 in maintaining RGL quiescence is in contrast with its role in progenitor proliferation during development of the cortex, which is conceptually important. These data reinforce that context matters.

1) A major conclusion hinges on the claim that it is Id4, not Id1-3, that inhibits ASCL1 protein accumulation by sequestering E47 heterodimerization partner. However much of the data does not directly exclude Id1-3. In fact, Id3 is expressed in some of the quiescent RGLs, albeit in a smaller proportion (~16.9%) and there is strong upregulation of Id3 in RGLs after Id4 deletion, suggesting the possibility that both Id3 and Id4 play redundant roles. Since ASCL1 is expressed in ~82.8% of quiescent RGLs, it is possible that it overlaps with Id3+ and Id4+ quiescent RGLs. Additional data addressing whether Id3 and Id4 are expressed in the same or different quiescent RGLs, together with ASCL1, is important to clarify this point. Also, direct evidence to exclude Id3's role in promoting the elimination of ASCL1 in quiescent RGLs will strengthen the claim that it is Id4 that maintains the quiescence of adult NSCs.

2) Throughout the paper, the quiescent state is defined by the lack of Ki67+ (Figure 4G, N; Figure 6G, H). Additional evidence supporting the induction of the quiescent state (for example, cell cycle analysis and quantification of cells in G1/G0 or complementary methods) would increase the rigor of the approach.

---

## [Author Response]

Summary:This manuscript adds important details to understanding the ways that dentate stem cells' quiescence is regulated. The reviewers were positive about the manuscript, but raised several issues that must be addressed in a revision of which the first point is the most critical. The original reviewers are included below.1) The role of other Id's. Is there redundancy or are there specific molecular functions of Id4. It is essential to include or exclude a role for Id1-3, given that the focus of the manuscript (and the title) is on Id4. (point 2, 3, 4 of reviewer 1, point 1 of reviewer 2).

We fully agree with you and the reviewers that this is the point needing clarification most. We have therefore performed new overexpression and knockdown experiments in NSC cultures as well as new RNA and immunolabelings in vivo to determine the importance of Id proteins other than Id4 for the degradation of Ascl1.

We have overexpressed Id1, Id2, Id3 and Id4 in proliferating adult NSCs in culture and examined the effect on Ascl1 protein levels (Figure 4—figure supplement 1E, subsection “Id4 promotes the degradation of Ascl1 protein and induces a quiescence-like state in NSCs”, first paragraph). The experiments were performed in parallel and using the same expression construct for all the Ids. It should be noted that Id2, Id3 and to a lesser extent Id1 are all expressed in proliferating adult neural stem cells, and colocalise with Ascl1. Confirming our previous results, we found reduced Ascl1 protein levels in cells overexpressing Id4. Acute overexpression of Id1, Id2 and Id3 was also sufficient to significantly reduce Ascl1 protein levels, suggesting that the other Ids can compensate for Id4 when expressed at high levels.

We have also performed knockdown experiments for the four Id genes in cultured NSCs in quiescent conditions, when all Id proteins are highly expressed and Ascl1 levels are very low (Figure 4—figure supplement 1F-H, subsection “Id4 promotes the degradation of Ascl1 protein and induces a quiescence-like state in NSCs”, second paragraph). We obtained a reasonably efficient knockdown of each Id gene individually. However, we did not observe a significant increase in Ascl1 protein levels with any Id protein knockdown. We also attempted a simultaneous knockdown of Id4 with Id1, 2 or 3, and again observed no significant increase in Ascl1 expression. One possible explanation arises from the observation that expression of Id1 and Id3 proteins increased following Id4 knockdown, suggesting that Id4 may normally suppress Id1 and Id3 expression. These results suggest that a simple knockdown of Id proteins in quiescent NSCs is confounded by the deregulation of the other Ids and their redundant activity of suppressing Ascl1 protein expression. The potential inhibition of Id1 and Id3 protein by Id4 and their increase upon Id4 knockdown may also explain the increase in Id1 and Id3 protein we observe in *Id4^cKO^* RGLs in vivo. These results also support the hypothesis that the other Id proteins, when expressed at high levels, can compensate for Id4.

Finally, our results using immunolabeling together with in situ hybridization by RNA Scope, shows that Id4 co-localises with Ascl1 mRNA in quiescent adult NSCs in vivo (Figure 3—figure supplement 2H-I, subsection “Id4 is highly expressed in quiescent hippocampal stem cells in culture and in vivo”, last paragraph). Id3 is expressed in a much smaller number of NSCs, and is expressed at similar levels in active (Ki67+) and quiescent (Ki67) NSCs in vivo, suggesting it has less of a role in maintaining NSC quiescence in vivo than Id4. Id3 also co-localises with Ascl1 mRNA in adult NSCs, although the vast majority of Id3+Ascl1+ NSCs co-express Id4. Id3 therefore could be involved in suppressing Ascl1 when Id4 is not present, although this represents a minority of NSCs (Figure 6—figure supplement 1J-M, subsection “Loss of Id4 in vivo activates quiescent adult hippocampal RGLs”, second paragraph).

2) The role of Hes family genes (point 1 of reviewer 1).

As reviewer 1 rightly points out, an increase in Hes proteins in response to increased Id4 expression could account for a reduction in Ascl1 mRNA levels and a subsequent reduction in Ascl1 protein levels. However, we do not observe a decrease in Ascl1 mRNA in cultured adult NSCs upon Id4 overexpression or when endogenous Id4 expression is high in quiescent conditions (previous Figures 2A-C, 4E and Figure 4—figure supplement 1H). In addition, we now show data from our RNA sequencing experiments demonstrating that Id4 overexpression does not increase the expression levels of Hes family genes (Hes1, Hes5 or Hey1) in cultured adult NSCs (Figure 5—figure supplement 1A, subsection “Quiescence is characterised by a downregulation of Ascl1 target genes”).

3) The cell cycle experiments would strengthen their conclusions but are non-essential (point 2 of reviewer 2).

We have performed a FACS-based cell cycle analysis in proliferating and quiescent conditions and found that indeed quiescent cultured adult NSCs are arrested in the G0/G1 phase (Figure 2—figure supplement 1D, subsection “Ascl1 is regulated post-translationally in quiescent NSC cultures”).

Reviewer #1:[…] There are several concerns in the functional analysis of Id4 and Ascl1.1) The authors indicated that Ids are upregulated upon BMP stimuli, and that Id4 negatively regulates the function of Ascl1 by destabilizing it. However, it is well-known that BMP also increases the expression of Hes family genes in some contexts, which could negatively regulate the function of Ascl1 and its expression. From their data that BMP stimulation does not decrease the mRNA levelS of Ascl1, it seems likely that Hes genes are not upregulated by BMP in their system, or that the expression of Hes does not influence the expression of Ascl1 in vitro. However, given all these caveats, it would be helpful if the authors examined the expression of Hes family genes in their qPCR or RNA-seq data.

We now show data from our RNA sequencing experiments demonstrating that BMP4 or Id4 overexpression does not increase the expression levels of Hes family genes (Hes1, Hes5 or Hey1) in cultured adult NSCs (Figure 5—figure supplement 1A, subsection “Quiescence is characterised by a downregulation of Ascl1 target genes”).

2) In Figure 6, they showed that Id1 and Id3 expression are upregulated in RGLs of Id4^cKO^ mice, supporting their idea that Id1 and Id3 compensate in part for Id4 function. However, the authors did not provide any data that Id1 and Id3 work similarly as Id4 for maintaining quiescence of NSCs and for the destabilization of Ascl1 (using their in vitro system). It is interesting to know if each Ids has different specific function or works similarly/together in NSCs. In this case, because BMP also upregulates the expression of Id1/3 not only Id4, I wonder if maintaining quiescent state and destabilization of Ascl1 is unique function of Id4, or if Id1/3 also can play similar role as Id4 in their culture system.

We have now performed overexpression and knockdown experiments of Id1 and Id3 and examined the effects on Ascl1 levels in cultured adult NSCs (Figures 4—figure supplement 1E and 1F-H, subsection “Id4 promotes the degradation of Ascl1 protein and induces a quiescence-like state in NSCs”). Our results show that, although Id3 and to a lesser extent Id1 are already highly expressed in proliferating NSCs, and are co-localized with Ascl1 protein (Figure 3—figure supplement 1B-I), over-expression of either Id1 or Id3 is sufficient to reduce Ascl1 protein levels. Knockdown of either Id1 or Id3 alone in quiescent NSCs is however not sufficient to affect Ascl1 levels. We also find that knockdown of Id4 results in an increase in Id1 and Id3 protein levels, and has little effect on Ascl1 protein levels, suggesting that deregulated Id1 and Id3 compensate the loss of Id4, and we propose that a similar compensatory mechanism occurs in *Id4^cKO^* mice. We have quantified the level of Id1 and, now, Id3 (Figure 3—figure supplement 2G) in active (Ki67+) and quiescent (Ki67-) NSCs in vivo, and found that Id3 is expressed at similar levels in both populations, whilst Id1 is enriched in active NSCs. We have now also shown that most Id3+ NSCs express Ascl1 mRNA but rarely without co-expression of Id4. These data support the hypothesis that homeostatic levels of Id1 is unlikely to regulate Ascl1 protein in NSCs in vivo, and that Id3 may contribute to the suppression of Ascl1 protein but at best in a very small number of NSCs. Id1 and Id3 can compensate for the loss of Id4 when expressed at high levels. These data suggest that, although all four Id proteins have the capacity to suppress Ascl1 protein expression when expressed at high levels in cultured NSCs, Id4 is the most important Id protein for maintaining low Ascl1 levels and quiescence of NSCs in vivo because of its expression pattern and levels in hippocampal NSCs.

3) The authors show that BMP signaling decreases the Ascl1 protein levels, and they suggested that it is mediated by Id4. To support this, the author showed that Id4 overexpression reduced Ascl1 protein amount by immunofluorescence staining. However, it is still possible that BMP signaling decreases Ascl1 protein using other mechanisms in addition to Id4. Given that, I wonder if they could perform some experiments to demonstrate how BMP signaling affects Ascl1 stability and NSC proliferation when Id4 protein is absent or reduced in their in vitro system. There are ways perhaps to simultaneously regulate Id1, 3 and 4 in this system.

We have not found a strategy to efficiently and simultaneously regulate the different Id proteins in our NSC culture system. However, as part of our parallel efforts in obtaining an Id4 conditional mutant line, we have obtained an adult hippocampal NSC line from Id4 floxed mice. Surprisingly, this line lacked Id4 expression altogether, even without any Cre recombinase expression, and we could not perform the planned acute deletion experiments (Author response image 1). We have decided not to include this data in the manuscript since we cannot explain the silencing of the Id4 floxed allele in cultured cells (Id4 floxed is expressed at normal levels in vivo, Author response image 1). Nevertheless, addition of BMP4 to this Id4-deficient cell line does not reduce the levels of Ascl1 protein, demonstrating that most if not all of the reduction of Ascl1 in quiescence conditions is due to the destabilizing effects of Id4. In addition, as mentioned above, we have now knocked down Id4 alone or in combination with Id1, Id2 or Id3 in quiescent culture conditions, and show that Id1 and Id3 may compensate for loss of Id4.

**Author response image 1. respfig1:** Id4 expression is absent in Id4floxed NSCs. (**A-B**) Immunolabeling for Id4, Ascl1 and DAPI staining in Id4^fl/fl^ NSCs in FGF2+BMP4 conditions, infected with either GFP-expressing or Cre-expressing adenovirus in order to delete Id4. No Id4 immunofluorescence was detectable in either control or Id4-deleted cells. Ascl1 levels were much high in both conditions, in contrast to wildtype cells in FGF2+BMP4 conditions. Scale bar, 30µm. (**C**) Quantification of Id4 expression levels by QPCR in wildtype (*wt*) control or Id4^fl/fl^ NSCs. Id4 expression is detected and increases in wildtype cells following BMP4 treatment, but is not detected or is detected at very low levels in Id4^fl/fl^ NSCs. n=1 for QPCR analysis; n=3 for RNAseq analysis.(**D**) Immunolabeling for Id1-4 in Id4^fl/fl^ NSCs in FGF2 and FGF2+BMP4 conditions. Id1-3 are all detected in Id4^fl/fl^ NSCs, but no signal for Id4 is detected. Scale bar, 30µm. (**E**) Immunolabeling for Id4 in *Glast^wt/wt^;Id4^fl/fl^;tdTomato*mice injected with tamoxifen for 5 days and analysed immediately. Id4 can be detected in many RGLs, suggesting they are not hypomorphic. Yellow arrows indicate Id4+tdtom+ RGLs. Scale bar, 30µm. Error bars represent mean ± SEM. Significance values: ns, p>0.05; *, p<0.05; **, p<0.01; ***, p<0.001; ****, p<0.0001.

4) They showed that Id4 competes with Ascl1 for binding with E47 in vitro. From this aspect, they assumed that the amount of Id4 increases monomeric of Ascl1 thereby destabilizing Ascl1. However, this mechanism isn't really clear. Furthermore, the reports they cited in the Discussion section showed that Id4 function is different from the other Ids. The binding affinity of Id4 to E47 is less than the one in other Ids (Sharma et al., 2015). This makes me wonder that destabilization of Ascl1 by Id4 is due to other mechanisms. The fact that deletion of Id4 increases the amount of Id1/3 without changing their protein level suggests that Id4 may also negatively regulate the protein stability of Id1/3. Because the paper also reported that Id4 makes heterodimer with other Ids to inhibit their function, I wonder that if they could also address the possibility that Id4 makes heterodimer with Ascl1 negatively regulates its function and leading to destabilization.

This is a tantalizing hypothesis. However, our co-IP experiments (Figure 3H) show no apparent binding between Ascl1 and Id4 in conditions in which both Ascl1 and Id4 do bind to Eeins.

Reviewer #2:[…] 1) A major conclusion hinges on the claim that it is Id4, not Id1-3, that inhibits ASCL1 protein accumulation by sequestering E47 heterodimerization partner. However much of the data does not directly exclude Id1-3. In fact, Id3 is expressed in some of the quiescent RGLs, albeit in a smaller proportion (~16.9%) and there is strong upregulation of Id3 in RGLs after Id4 deletion, suggesting the possibility that both Id3 and Id4 play redundant roles. Since ASCL1 is expressed in ~82.8% of quiescent RGLs, it is possible that it overlaps with Id3+ and Id4+ quiescent RGLs. Additional data addressing whether Id3 and Id4 are expressed in the same or different quiescent RGLs, together with ASCL1, is important to clarify this point. Also, direct evidence to exclude Id3's role in promoting the elimination of ASCL1 in quiescent RGLs will strengthen the claim that it is Id4 that maintains the quiescence of adult NSCs.

We now show immunostaining and in situ hybridization by RNA Scope in vivo showing that Id4 is expressed by most quiescence adult NSCs and co-localises with Ascl1 mRNA. Id3 is also expressed in subsets of both quiescent and active NSCs, and often co-localises with Id4 in control mice. We do hypothesize that Id3 might play a role in regulating Ascl1 protein levels in the few NSCs where Id4 is absent. To better define the role of the different Id proteins in the regulation of Ascl1 stability, we have performed new overexpression and knockdown experiments using our NSC culture system as discussed in response to point 2 of reviewer 1 (Figure 4—figure supplement 1E and 1F-H, subsection “Id4 promotes the degradation of Ascl1 protein and induces a quiescence-like state in NSCs”).

2) Throughout the paper, the quiescent state is defined by the lack of Ki67+ (Figure 4G, N; Figure 6G, H). Additional evidence supporting the induction of the quiescent state (for example, cell cycle analysis and quantification of cells in G1/G0 or complementary methods) would increase the rigor of the approach.

We have now performed a FACS-based cell cycle analysis in proliferating and quiescent conditions and found that indeed quiescent adult neural stem cells are arrested in the G0/G1 phase (Figure 2—figure supplement 1D, subsection “Ascl1 is regulated post-translationally in quiescent NSC cultures”).